# Real-space recipes for general topological crystalline states

Zhida Song[1,2,9], Chen Fang [1,3,4,5 ✉] & Yang Qi [6,7,8 ✉]

Topological crystalline states (TCSs) are short-range entangled states jointly protected by onsite and crystalline symmetries. Here we present a unified scheme for constructing all TCSs, bosonic and fermionic, free and interacting, from real-space building blocks and connectors. Building blocks are lower-dimensional topological states protected by onsite symmetries alone, and connectors are glues that complete the open edges shared by two or multiple building blocks. The resulted assemblies are selected against two physical criteria we call the no-open-edge condition and the bubble equivalence. The scheme is then applied to obtaining the full classification of bosonic TCSs protected by several onsite symmetry groups and each of the 17 wallpaper groups in two dimensions and 230 space groups in three dimensions. We claim that our construction scheme can give the complete set of TCSs for bosons and fermions, and prove the boson case analytically using a spectral-sequence expansion.

[1] Beijing National Research Center for Condensed Matter Physics, and Institute of Physics, Chinese Academy of Sciences, 100190 Beijing, China. [2] University of Chinese Academy of Sciences, 100049 Beijing, China. [3] Kavli Institute for Theoretical Sciences, Chinese Academy of Sciences, 100190 Beijing, China. [4] CAS Center for Excellence in Topological Quantum Computation, Beijing, China. [5] Songshan Lake Laboratory For Materials Science, 523808 Guangdong, China. [6] State Key Laboratory of Surface Physics, Fudan University, 200433 Shanghai, China. [7] Center for Field Theory and Particle Physics, Department of Physics, Fudan University, 200433 Shanghai, China. [8] Collaborative Innovation Center of Advanced Microstructures, 210093 Nanjing, China. [9] Present address: Department of Physics, Princeton University, Princeton, NJ 08544, USA. ✉email: cfang@iphy.ac.cn; qiyang@fudan.edu.cn

Symmetry-protected topological states (SPT)[1–4] are gapped states that do not have topological orders[5,6] (fractional excitations), but cannot be deformed into product states of localized wave functions without either symmetry breaking or gap closing. The constituent particles of SPT can either be bosonic or fermionic. They are probably the most well-understood topological states, and the famous examples are AKLT-like states[7] (bosonic), topological insulators, and topological superconductors[8,9] (both fermionic). Especially, SPT protected by onsite symmetries (only acting on internal degrees of freedom) have been studied for years, and we now know that bosonic SPT is classified by group cohomology of the symmetry group[1–4,10] (with the exception of the so-called "beyond-group-cohomology" states[11–14]), and SPT of free fermions is classified by the K theory in the "tenfold way"[15,16]. The classification of interacting fermions is much harder. Progresses in recent years[17–27] have provided mathematical frameworks to describe the classification, but the detailed computation is still challenging for general symmetry groups. In contrast to SPT protected by onsite symmetries are crystalline symmetry-protected topological states, or simply topological crystalline states (TCS).

As suggested by name, TCS has its nontrivial topology protected by both onsite and crystalline symmetries. Crystalline symmetries are the symmetry groups of periodic lattices in various dimensions (restricted, for simplicity, to two and three in this paper), and the study of crystalline symmetries as groups has been complete since the end of the last century[28]. All crystalline symmetries are classified into 17 wallpaper groups into two dimensions (2D) and 230 space groups into three dimensions (3D). Among TCS, those constituted of noninteracting fermions with charge conservation have so far attracted most theoretical and experimental effort. These states are also known as the topological crystalline insulators[29,30], the classification and diagnosis of which have only recently been completed[31–36]. Interacting TCS, especially the fermionic one, is far less understood. On one hand, the framework of group cohomology for onsite-symmetry bosonic SPT cannot be directly applied; on the other hand, there is not an obvious way of adapting the K theory, which is key to solving the classification problem of free fermions[16,37,38], to the task of classifying interaction fermions. A recent work by Thorngren and Else[39] provides a mathematical connection between TCS and onsite-symmetry SPT states. Another way to understand TCS is the process of dimensional reduction[40–42], which constructs TCS by decorating high-symmetry points, lines, and planes with lower-dimensional onsite-symmetry SPT. These dimensional-reduction constructions are easier to compute in practice (because both the dimensionality and symmetry groups are reduced), and offer extra physical insight into the nature of these TCS states. For interacting bosonic and fermionic SPT states, a large class of TCS states has been constructed this way, but the previous works have not been systematically extended to arbitrary symmetry groups, and it has not been shown whether the real-space construction is complete.

We in this paper show that all TCS, bosonic and fermionic, free and interacting, can be built up in real space from two types of elementary ingredients, the building blocks (or simply blocks) and the connectors. Building blocks are finite-size pieces of lower-dimensional SPT that are protected by the respective little symmetry group alone. The little symmetry group includes all symmetry operations, onsite or spatial, that leave each point in the lower-dimensional subspace invariant: they can be considered as enlarged onsite groups by the spatial symmetries that do not change the spatial coordinates on specific subspaces of the lattice. A building block defined on a $p$-dimensional subspace is called a $p$ block. For three-dimensional TCS, one considers $p = 3, 2, 1, 0$ blocks (3-block TCSs are simply 3D SPT states protected by the

onsite symmetry alone, which are compatible with the crystalline symmetries. This is discussed in more detail in Sec. V of Supplementary Information). To construct a gapped TCS, we arrange $p$ blocks in such a way that the space group of the TCS is preserved, including translation symmetries, point-group symmetries, and nonsymmorphic symmetries. The $p$ blocks in general have open boundaries, and, being SPT themselves, gapless boundary states (or more precisely speaking, boundary anomalies) on their $(p − 1)$-dimensional boundaries. Therefore, a symmetric construction with $p$ blocks alone cannot be gapped in the bulk, and in order to build a gapped state, one needs "glue" to close the open edges in the assembly. The glue is the connector, which, technically speaking, is a torsor defined on $(p − 1)$ dimensions. $(p − 1)$ connectors are inserted where multiple $p$ blocks share one $(p − 1)$-dimensional open edge, and should hybridize the gapless states contributed from the joining $p$ blocks so that the edge becomes gapped. When all open edges are completed, that is, when the "no-open-edge condition" is met, we obtain assemblies that are (i) symmetric under onsite and spatial symmetries and (ii) gapped. However, this does not mean that the crystal is topologically nontrivial, as we additionally require that it cannot be deformed into a product state. Obviously, this implies that there is at least one building block with $p > 0$ that is a nontrivial SPT, but this alone is insufficient: there are constructions from nontrivial $(p > 0)$-building blocks that can still be trivialized. We show that the trivialization can be understood as a "bubbling process," in which constructions that can be canceled by the "bubbles" are excluded, considered trivial. Two TCS are hence topologically equivalent if the decorations can be related by a bubbling process, and this is called the "bubble equivalence". The space of all TCS is hence the space of symmetric assemblies of building blocks satisfying the no-open-edge condition quotient of the bubble equivalence.

One should be aware that both the no-open-edge condition and the bubbling equivalence, simple enough in appearance, have their subtleties. While it is obvious that one may use a $(p − 1)$ connector to complete the open edges at the meeting of two or multiple $p$ blocks, after all necessary $(p − 1)$ connectors are added, at the $(p − 2)$ joints where these $(p − 1)$ connectors meet, there may be $(p − 2)$-dimensional open edges. Similarly, while it is natural that bubbles in $p + 1$ dimensions can be used to annihilate $p$ blocks, there are cases where $(p + 2)$ bubbles, leaving all $(p + 1)$ blocks intact, annihilate $p$ blocks. A third and related subtlety, called the group-extension problem, is about the relations between TCS constructed from $p$ blocks and those constructed from $(p' < p)$ blocks. All three subtleties have to do with constructions that have nontrivial connectors or torsors. Torsors are not SPT (but may be understood as fractions of SPT), and their topological properties should be separately considered.

The real-space construction scheme given above allows an automated generation of all inequivalent TCS for arbitrary spatial and onsite symmetry groups in any dimension $D$, defined in the following steps: (a) make a symmetric arrangement of $p$ blocks, (b) add $(p − 1)$ connectors to complete open edges, (c) use $(p + 1)$ bubbles to "modulo out" trivial constructions, (d) add $(p − 2)$ connectors to complete open edges of $(p − 1)$ connectors, (e) use $p + 2$ bubbles to "modulo out" trivial constructions, and (f) repeat until the connectors are zero-dimensional and the bubbles are $d$-dimensional, where $d$ is the spatial dimension. This process naturally fits the construction of TCS into a spectral sequence, a successive approximation originally designed for computing homology (cohomology) groups of a topological space[43,44]. We adapt the terminology in mathematics and refer to different orders in this successive approximation as different pages in the spectral sequence. Each page, being worked out from the previous page, can be roughly understood as a certain level of

approximation to the exact classification, more accurate than its previous page, and less accurate than the next. Following this observation, we can analytically prove that the real-space construction process as defined above gives exactly the same classification of general bosonic TCS, as derived from the gauging-spatial-symmetry argument[34]. This proof is presented in Sec. IV of the Supplementary Information. We develop an automated code and generate all bosonic TCS having typical onsite symmetries (such as unitary and antiunitary $Z_n$ symmetries), and directly multiply any of the 17 wallpaper groups in 2D and 230 space groups in 3D (see Sec. VII of Supplementary Information). On the fermionic side, the completeness of the construction scheme has been demonstrated for free fermions with charge conservation, time-reversal symmetry, and any of the 230 space group symmetries in ref. [36]. We also point out that while the general real-space construction still holds, the difficulty in classifying interacting fermionic TCS lies in the nontrivial superposition of states due to fermion statistics, and the lack of a unified bulk and boundary description for fermionic SPT.

Compared with the group cohomology formula in ref. [39], our method for classifying TCS is not only easier to compute, but also more concrete so that for each TCS, we have a real-space, piecewise construction. In particular, it allows us to distinguish TCS made of building blocks in different dimensions, which is an additional structure in the TCS classification. In addition, the paginated structure of the spectral sequence also has physical interpretations: the different levels of approximation can be used to construct variants of the Hasting–Oshikawa–Lieb–Schultz–Mattis (HOLSM) Theorem[42,45–55].

## Results

**Cell decomposition and chain complex.** Following ref. [36], for each space group, there is a well-defined cell decomposition process that a $d$-dimensional Euclidean space ($R^d$) is decomposed into the union of $p = 0, 1, 2, \ldots, d$-dimensional "cells" with zero overlap. Here a $p$ cell is topologically equivalent to $R^p$, or a $p$-dimensional disk minus its boundary, and we denote the collection of all $p$ cells in a decomposition as $Y_p$. We use Greek letters $\sigma$, $\tau$, $\mu$, and $\gamma$ to denote $p = d$, $d-1$, $d-2$, $d-3$-cells, respectively, and also use $\sigma$ to denote a general cell with unspecified dimension. We require that any space-group operation maps one $p$ cell to another $p$ cell. We also require that the union of all $Y_p$ is $R^d$ itself. Mathematically, the collection of all cells under these conditions forms a topological space, $Y$, called a $G$ complex. Furthermore, for any $\sigma \in Y_p$, we consider the subgroup $G_\sigma$ that maps $\sigma$ to itself. We require that any $g \in G_\sigma$ has a pointwise action on $\sigma$, which in general includes onsite and crystalline symmetries. In other words, $G_\sigma$ acts as an onsite symmetry group locally on $\sigma$. In addition, unlike in ref. [36], the cells are oriented. The orientation is arbitrarily subject to the condition that the orientations of $\sigma$ and $g \cdot \sigma$ are also related by $g$.

We illustrate the cell decomposition with simple examples. In Fig. 1a, we show the cell decomposition within one unit cell of 2D space group $pm$, having orthogonal basis vectors with one mirror line mapping $x$ to $-x$. In Fig. 1b, there is another example of decomposition for 2D space group $p2$, having a twofold rotation center $C_2$. For this decomposition, we comment that the 1D segment passing the rotation center, instead of being a 1-cell by itself, is decomposed into two 1-cell ($\tau_3$ and $C_2\tau_3$) and one 0-cell ($\mu_4$). This is in contrast to the example in Fig. 1a, where the segment coincident with the mirror line needs no additional decomposition. The difference is because we require that symmetry group of a $p$ cell, $G_\sigma$, should act pointwise on $\sigma$, and while each point in $\tau_3$ is invariant under a mirror in Fig. 1a, only $\mu_4$ is invariant under $C_2$, and $\tau_3$ and $C_2\tau_3$ are mapped to each other.

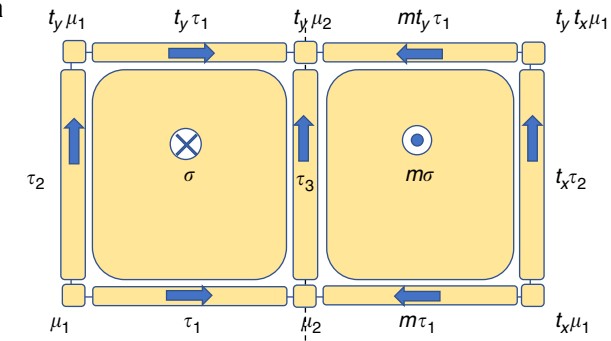

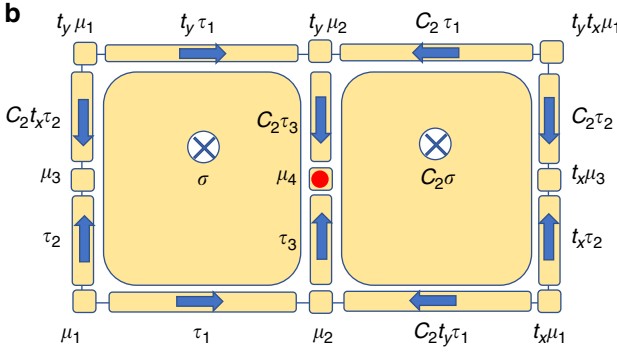

**Fig. 1 Cell decomposition for 2D wallpaper groups. a** The cell decomposition for 2D wallpaper group $Pm$. The mirror line is the vertical dashed line. **b** The cell decomposition for 2D wallpaper group $P2$. The rotation center is marked in red. In both panels, the independent cells are labeled by $\sigma_{1,2,3,\ldots}$ ($\tau$ for 1-cell and $\mu$ for 0-cell), and the other cells are labeled as $g\sigma$ with $g \in G$. Their orientations are marked by arrows.

After decomposition, we are ready to define the operator $\partial$ acting on a $p$ cell $\sigma_p$: $\partial\sigma_p$ gives the $(p-1)$ cells that are at the boundary at $\sigma$, i.e.,

$$\partial\sigma_p = \sum_{\sigma_{p-1} \subset \sigma_p} \langle \sigma_{p-1} | \partial\sigma_p \rangle \sigma_{p-1}. \tag{1}$$

The coefficient $\langle \sigma_{p-1} | \partial\sigma_p \rangle = 0$, if $\sigma_{p-1}$ is not part of the boundary of $\sigma_p$; $\langle \sigma_{p-1} | \partial\sigma_p \rangle = \pm 1$, if $\sigma_{p-1}$ is part of the boundary, and the sign depends on the relative orientations of $\sigma_p$ and $\sigma_{p-1}$. For example, in Fig. 1a, the orientation of $\sigma$ gives, by right-hand rule, natural orientations for adjacent 1-cell, and if the designated orientation of $\tau_i$ is parallel (opposite) to it, $\langle \tau_i | \partial\sigma \rangle = +1$ ($-1$): $\partial\sigma = \tau_1 + \tau_3 - t_y\tau_1 - \tau_2$. For 1-cell $\tau_i$, the arrow naturally gives the coefficients $\langle \mu_i | \partial\tau_j \rangle$: the zero cell at the head and at the tail of the arrow has $+1$ and $-1$ coefficients, respectively. For example, $\partial\tau_1 = \mu_2 - \mu_1$. The $\partial$ operation hence defines a chain complex from $d$ cells all the way down to 0-cell:

$$Y_d \xrightarrow{\partial} Y_{d-1} \xrightarrow{\partial} \cdots \xrightarrow{\partial} Y_0. \tag{2}$$

Here, we notice that the homology groups of this chain complex are all trivial, because $Y$ is a decomposition of $R^d$, which is topologically trivial. As a result, the chain complex in Eq. (2) must be an exact sequence, i.e., $\partial^2 = 0$, which is important to the proof in Sec. IV of Supplementary Information.

**Building blocks and connectors.** We start by reviewing topological states we can decorate on each cell. This forms the basis of our real-space construction: each cell $\sigma$ of dimension $p$ can be decorated with a topological state, which is protected by the local onsite symmetry group $G_\sigma$. On an isolated cell, such topological

states are classified by the $p$-dimensional SPT states protected by $G_\sigma$, which we denote by $\Phi^p(G_\sigma)$. We will refer to such decorations as "$p$-dimensional building blocks", or $p$ blocks for short. However, in our real-space construction, we need to consider a more general case, where $\sigma$ is part of the boundary of a $(p+1)$-dimensional cell $\tau$, which is in turn decorated by another topological state. Hence, we need to extend the scope of topological states to also include boundary topological states. In general, we consider a set of representative fixed-point topological states, which we denote by $\Psi^p(G)$.

There are two important operations on $\Psi^p(G)$, which are the mathematical foundation for the computation of our real-space constructions. First, we can stack two states $\psi_{1,2}$ in $\Psi^p(G)$, and get a new state, which we denote by $\psi_1 \boxplus \psi_2$. Second, we can compute the coboundary of $\psi \in \Psi^p(G)$, which should be a state in $\Psi^{p+1}(G)$. The physical meaning of $d\psi$ is that $\psi$ can be realized on a $p$-dimensional boundary of $d\psi$.

The structure of such sets $\Psi^p(G)$ and the rules of these two operations depend on the nature of the system: being a free-fermion, interacting-boson, or interacting-fermion system. In particular, for bosonic systems, $\Psi^p(G)$ is simply the cochain space of $G$. In the main text, we assume that $\Psi^p(G)$, $\boxplus$ and $d$ are known, and use them to compute real-space constructions, while leaving the details to Sec. I of Supplementary Information. In particular, the SPT phases $\Phi^p(G)$ can be computed from $\Psi^p(G)$, and the coboundary maps $d: \Phi^p(G)$ are given by the subgroup of cochains in $\Psi^p(G)$ satisfying $d\alpha = 0$, quotient of the coboundaries given by the image of the coboundary map $d[\Psi^{p-1}(G)]$.

**"First-page" (first-order approximation) candidates for TCS.** In general, a $d$-dimensional TCS can be constructed by assembling $p$ blocks $(p < d)$ and gluing them with lower-dimensional connectors, while leaving all higher-dimensional cells empty. The state after this construction is termed a $p$-block topological crystal[36,41,42]. In this way, all TCSs are organized into $p$-block topological crystals, where $p = 0, \ldots, d$: in particular, one can argue that a TCS with trivial (but not necessarily empty) SPT states decorated on $d > p$ cells can be continuously deformed to a topological crystal with all $(d > p)$ cells empty[36]. The classification and construction of TCS then amounts to enumerating all inequivalent topological crystals for a given symmetry group $G$. Here, when enumerating all $p$-block topological crystals, we do not distinguish different choices of lower-dimensional connectors. This is because the difference (the difference between two states $A$ and $B$ is defined as the stacking of $A$ and the inverted state of $B$, denoted by $-B$) between two such choices on $q$-cell connectors can be viewed as a $q$-block topological crystal and described by the $q$-block classification.

We now consider $p$-block topological crystals. They are labeled by different choices of $p$ blocks, which are $p$-dimensional SPT states at $p$ cells $\sigma \in Y_p$. Their classification is determined by the local onsite group $G_\sigma$, and given by $\Phi^p(G_\sigma)$. For bosonic systems, there is $\Phi^p(G_\sigma) = H^{p+1}[G_\sigma, U(1)]$. However, for interacting fermionic systems, the calculation of $\Phi^p(G_\sigma)$ for arbitrary $G_\sigma$ can be much more complicated[17–27].

Naturally, if some $p$ cell $\sigma$ is decorated with some $p$ block, $[\alpha]$, then symmetry requires that the $p$ cell $g\sigma$ must be decorated by the $p$ block $g \cdot [\alpha]_\sigma \in \Phi_{g\sigma}$ for $g \in G$. This is possible because of the isomorphism $G_{g\sigma} = gG_\sigma g^{-1} \simeq G_\sigma$. In Sec. II of Supplementary Information, we show the explicit definition of $g \cdot [\alpha]$, and for now, we intuitively understand it as copying the $p$ block at $p$ cell $\sigma$ to another $p$ cell $g\sigma$. Therefore, only the cells in the $G$ orbits of $Y_p$, $Y_p/G$, may have independent $p$ blocks, and the $p$ blocks at all the other cells are determined by symmetry. Physically, $\sigma \in Y_p/G$ are the $p$ cells that are not related by any $g \in G$ and $g \neq I$. Once the

$p$ blocks for all cells in $Y_p/G$ are specified, we obtain a symmetric assembly denoted by $[\psi_p]$, such that

$$[\psi_p|_\sigma] \in \Phi^p(G_\sigma) \tag{3}$$

and

$$[\psi_p|_{g\sigma}] = g \cdot [\psi_p|_\sigma]. \tag{4}$$

Throughout the paper, we use $\psi_p$ to denote the state, or wave function, on all $p$ cells, use $|_\sigma$ to denote the same wave function restricted to a certain cell $\sigma$. We use $[\psi]$ for the phase, the representative state of which is $\psi$. The collection of all possible symmetric assemblies from $p$ blocks is denoted by $E^p_{p,1}$

$$E^p_{p,1} \equiv \{[\psi_p]\} \equiv \bigoplus_{\sigma \in Y_p/G} \Phi^p(G_\sigma). \tag{5}$$

For reasons to be revealed in later sections, these assemblies are called the first-page candidates for TCS, which can also be taken as first-order approximations to TCS. The exact meaning of $E^q_{p,r}$ will also be introduced and put to use in due course.

**No-open-edge condition.** Being SPT, $[\psi_\sigma]$ necessarily has gapless modes at $\tau \in \partial\sigma$. In fact, in the $G$ complex, each $(p-1)$ cell is in the boundary of at least two $p$ cells. (For example, $\tau_3 \in \partial\sigma \cap \partial m\sigma$ in Fig. 1a.) Therefore, for a gapped TCS, we require that the gapless modes at $\sigma$ contributed from all adjacent $\tau$ can gap out each other, so that there is no open edge. More precisely speaking, the gapless boundary modes reflect the quantum anomaly of an SPT boundary. The necessary condition for a gapped $(p-1)$ cell $\tau$ is that the anomalies contributed by all adjacent $[\psi_p|_\sigma]$ cancel each other.

Suppose a $(p-1)$ cell $\tau$ is the common edge of $n$ pieces of $p$ cells $\sigma_{1,\ldots,n}$, each decorated by a $p$ cell $[\psi_p|_{\sigma_i}] \in \Phi^p(G_{\sigma_i})$. Then we define the following quantity for $\tau \in Y_{p-1}$, given any $\psi_p$:

$$[d_1\psi_p|_\tau] \equiv \bigoplus_{\sigma \in Y_p} \langle \tau|\partial\sigma \rangle [\psi_p|_\sigma] \in \Phi^p(G_\tau). \tag{6}$$

Physically, $[d_1\psi_p|_\tau]$ is an SPT obtained from stacking $[\psi_p|_\sigma]$ for $\langle \tau|\partial\sigma \rangle \neq 0$, and is a $p$-SPT with symmetry group $G_\tau \supseteq G_\sigma$. The collection of this $p$-SPT attached to $(p-1)$ cells is denoted by $E^p_{p-1,1}$

$$E^p_{p-1,1} \equiv \bigoplus_{\sigma \in Y_{p-1}/G} \Phi^p(G_\sigma). \tag{7}$$

If $[d_1\psi_p|_\tau]$ is a nontrivial SPT with respect to $G_\tau$, then on $\tau$, which is the common edge of the stacked layers, there must be gapless modes, meaning that this particular $\psi_p$ fails the no-open-edge condition. Hence, the nonzero elements in $E^p_{p-1,1}$ are also called anomaly patterns. For a given $[\psi_p] \in E^p_{p,1}$, the collection of all $[d_1\psi_p|_\tau]$ is given by $d_1[\psi_p] = \oplus_{\tau \in Y_{p-1}}[d_1\psi_p|_\tau]$. This defines $d_1$ as a linear mapping between $E^p_{p,1}$ to $E^p_{p-1,1}$. Therefore, the no-open-edge condition states that only the kernel of $d_1$ may be candidates for TCS, i.e., $d_1[\psi_p] = 0$.

On the other hand, if $[d_1\psi_p|_\tau]$ is a trivial $p$-SPT with symmetry group $G_\tau$, on $\tau$ we can place a "mass term" that gaps out the edge modes contributed by $\sigma_{1,\ldots,n}$. The resulted gapped state on the $(p-1)$ cell $\tau$ is called a connector, as in real space it acts as the nexus of the bordering $p$ cells $\sigma_{1,\ldots,n}$. Connectors are not SPT in general, but are torsors (see "Methods"). For now, we only need to know that any $[\psi_p] \in E^p_{p,1}$ that satisfies the no-open-edge condition can be glued by $(p-1)$ connectors such that any $\tau \in Y_{p-1}$ is also gapped.

**Bubble equivalence**. The kernel of $d_1 : E^p_{p,1} \to E^p_{p-1,1}$ are gapped, symmetric assemblies of $p$ blocks (gapped at any $\tau \in Y_{p-1}$ by connectors). But two different assemblies in $\ker d_1$ may be topologically equivalent to each other, and more importantly, some nontrivial (nonvacuum) state in $\ker d_1$ may even be equivalent to vacuum upon adiabatic deformation. Following ref. [36], every adiabatic deformation is equivalent to the creation (annihilation) of bubbles within some $p$ cell ($0 < p \le d$). A $p$ bubble is a $p$-dimensional disk inside some $p$-cell $\nu \in Y_p$, the inside of which is vacuum, and the boundary of some $(p-1)$-SPT protected by $G_\nu$. Therefore, it can be considered as some $(p-1)$-SPT attached to a $p$ cell. It is straightforward to see that creation of bubbles cannot change the topology of the state. Any bubble can shrink to a point and vanish, and, as the inside of any cell in our cell decomposition does not have any spatial symmetry, the shrinking and vanishing process does not break any symmetry.

Therefore, the topology of $[\psi_p]$ is unchanged after we decorate, in a $G$-symmetric way, the $(p+1)$ cells with some $(p+1)$ bubbles. A $G$-symmetric assembly of $(p+1)$ bubbles is denoted $[\tilde{\psi}_{p+1}]$ and $[\tilde{\psi}_{p+1}]|_\nu$ is a part of the assembly restricted to some $\nu \in Y_{p+1}$. The collection of all $(p+1)$ bubbles is denoted by $E^p_{p+1,1}$, defined in a way similar to Eq. (5):

$$E^{p+1}_{p,1} \equiv \bigoplus_{\sigma \in Y_p/G} \Phi^{p+1}(G_\sigma). \tag{8}$$

We refer to $[\tilde{\psi}_{p+1}] \in E^{p+1}_{p,1}$ as a bubbling pattern.

The equivalence relations induced by bubbles in $E^{p+1}_{p,1}$ are expressed by another linear map $\tilde{d}_1$, which has a form similar to Eq. (6),

$$\left[\tilde{d}_1 \tilde{\psi}_{p+1}\Big|_\sigma\right] \equiv \bigoplus_{\nu \in Y_{p+1}} \langle \sigma | \partial \nu \rangle [\tilde{\psi}_{p+1}|_\nu] \in \Phi^p(G_\sigma),$$
$$\tilde{d}_1[\tilde{\psi}_{p+1}] = \sum_{\sigma \in Y_p} [\tilde{d}_1 \tilde{\psi}_{p+1}|_\sigma]. \tag{9}$$

As shown in the section "Bubbling equivalence on the first page", $\tilde{d}_1 \tilde{\psi}_{p+1}$ gives the state generated by deforming the bubbling pattern $\tilde{\psi}_{p+1}$, which should be regarded as a trivial state. These trivial states in $\mathrm{img} \tilde{d}_1$ give the equivalence relations in $E^p_{p,1}$: $[\psi_p] \sim [\psi'_p]$, if and only if $[\psi_p] - [\psi'_p] \in \mathrm{img}\, \tilde{d}_1$.

**"Second-page" candidates for TCS**. The no-open-edge condition requires that TCS be in the kernel of $d_1$ in Eq. (6), and the bubble equivalence states that two TCS related by an image of $\tilde{d}_1$ are topologically the same. The first-order approximation to TCS, $E^p_{p,1}$, is refined by these two conditions into

$$E^p_{p,2} \equiv \frac{\ker d^p_{p,1}}{\mathrm{img}\, d^p_{p+1,1}}, \tag{10}$$

where we introduce a general version of $d_1$ and $\tilde{d}_1$: $d^p_{q,r} : E^p_{q,r} \to E^{p+1-r}_{q-r,r}$. $d_1$ and $\tilde{d}_1$ are the special cases $d_1 = d^p_{p-1,1}$, $\tilde{d}_1 = d^p_{p+1,1}$.

Like those in $E^p_{p,1}$, states in $E^p_{p,2}$ are also generated by decorating $p$ cells and are $G$-symmetric, but no-open-edge condition ensures that all cells in $Y_{p-1}$ are gapped, and the bubble equivalence relation ensures that they cannot be trivialized by $(p+1)$ bubbles. $E^p_{p,2}$ is hence the second-page approximation to the set of TCS made from $p$ blocks ($p$-TCS for short). Here, page is the mathematical terminology referring to the order of approximation in the spectral sequence, and we denote the page number by $r$ in the notation $E^q_{p,r}$. In further sections, we are to treat higher-page approximations $E^p_{p,3}, E^p_{p,4}...$, and in the end $E^p_{p,\infty}$ is the exact collection of $p$-TCS. In Sec. V of Supplementary Information,

however, we show that for bosonic systems, if $G$ is a direct product of the onsite and the space symmetry groups $G = SG \otimes G_0$, then the second page already contains the final answer: $E^p_{p,2} = E^p_{p,\infty}$, which can be simply summed to produce the full classification

$$\bigoplus_{p=1}^d E^p_{p,\infty} = \bigoplus_{p=1}^d E^p_{p,2}. \tag{11}$$

We remark that in this work, the SPT with 0-block is considered trivial, hence the lower bound in the summation in Eq. (11).

**Further considerations**. The discussion presented so far provides an algorithm to enumerate possible topological crystals from lower-dimensional building blocks. In this algorithm, we start by decorating $p$-dimensional cells with SPT states protected by the local symmetry groups. We then compute whether the resulting anomalies cancel on the $(p-1)$ cells, and whether the state can be trivialized by a trivialization pattern on the $(p+1)$ cells. However, a careful mathematical analysis in the section "Connectors and second-page results" reveals that, for a general symmetry group $G$, such an intuitive algorithm is not complete as the answers in $E^p_{p,2}$ contain false entries that do not represent valid and nontrivial SPT states.

On one hand, $E^p_{p,2}$ may contain invalid entries: although we have checked that the building blocks can be glued together in a gapped way along the $(p-1)$-dimensional edges, it is possible that doing so will always leave gapless modes on the $(p-2)$-dimensional cells. In fact, two examples will be given in the section "Second-page no-open-edge conditions". Therefore, to make sure that an SPT building block represents a gapped SPT state, one needs to check that if the decoration can be extended to all lower-dimensional cells without anomaly.

On the other hand, $E^p_{p,2}$ may contain trivial entries that can be revealed by considering higher-dimensional bubbles. It is possible that bubbles on $(p+2)$-dimensional cells do not create any nontrivial SPT decorations on the $(p+1)$ cells, but create such decorations on the $p$ cells. Such a pattern of bubbles then indicates that certain decoration on $p$ cells, $[\psi_p] \in E^p_{p,2}$ it creates is trivial. In fact, such an example will be given in the section "Application to HOLSM Theorems". Consequently, to eliminate all trivial entries, one has to consider bubbles on all higher-dimensional cells, bounded by $d$, the spatial dimension.

There is one more subtlety we need to consider. We have generally divided TCS according to the dimensionality of the building blocks. For example, the 3D TCS can be divided into SPTs with plane decorations and line decorations, represented by $E^2_{2,\infty}$ and $E^1_{1,\infty}$, respectively. (Point decorations correspond to atomic insulators and do not have boundary states in general, and so are excluded from the set of TCS. Nevertheless, they can also be easily enumerated using our classification scheme.) The complete classification of TCS is then a combination of all topological crystals with all possible $0 < p < d$. However, when recombining the classification of topological crystals with different building-block dimensions, the result may be a nontrivial group extension of the two respective groups, instead of a direct sum. For example, assume that for some $G$, $E^2_{2,\infty}$ and $E^1_{1,\infty}$ are both $\mathbb{Z}_2$. Naively, the combined classification would be $\mathbb{Z}_2 \times \mathbb{Z}_2$. However, the correct result could also be $\mathbb{Z}_4$, which is a nontrivial extension of $\mathbb{Z}_2$ over $\mathbb{Z}_2$. Intuitively, imagine stacking two copies of the nontrivial elements from $E^2_{2,\infty}$. Since the classification is $\mathbb{Z}_2$, the resulting SPT state is trivial if viewed as a

topological crystal with $p = 2$, i.e., the decoration on the 2-cells is trivial. However, the resulting state can have nontrivial decorations on the 1-cell, and thus is the nontrivial 1-TCS, or, the generator of $E_{1,\infty}^1$. If this happens, the combined classification is then $\mathbb{Z}_4$, generated by the generator of $E_{2,\infty}^2$. For example, we consider a free fermion system with the time-reversal symmetry and the inversion symmetry. The classifications protected by the time-reversal symmetry (the local symmetry) in 3D and 2D are both $\mathbb{Z}_2$. However, the combined classification is $\mathbb{Z}_4$. The stack of two identical 3D centrosymmetric $\mathbb{Z}_2$ topological states is a topological state jointly protected by the time-reversal and inversion symmetries, and is equivalent to a construction of 2D $\mathbb{Z}_2$ topological states in certain 2-cells[35,36,56]. In general, stacking two or multiple $p$-TCS may produce trivial decorations on $p$ cells, but nontrivial decorations on $p'$ cells where $p'$, which is a nontrivial $p'$-TCS. Finding these relations in general is what we call a group-extension problem. All the three subtleties outlined above involve the connectors on lower-dimensional cells, which are discussed in the section "Connectors and second-page results".

**Application to HOLSM theorems**. Our TCS constructions can be applied to obtain generalized HOLSM-type theorems[45–52], and the SPT-enforcing HOLSM Theorems[53–55,57]. Using the higher-page derivatives introduced in the section "Connectors and second-page results", these HOLSM-like constraints can be summarized as the following in our language: a spatial distribution of projective representations, such as the Kramers doublets in the original HOLSM, is represented by an anomaly pattern $[\tilde{\psi}]$ in $E_{0,1}^1$. If $[\tilde{\psi}]$ is trivialized through $d_r$ by a $[\psi] \in E_{r,r}^r$ as $[\tilde{\psi}] = d_r[\psi]$, then $[\tilde{\psi}]$ can be gapped out by the r-block TCS assembly $[\psi]$. Otherwise, if $[\tilde{\psi}]$ remains a nontrivial element in $E_{0,\infty}^1$, then it cannot be gapped out by an SPT state, and will give the consequences of the original LSM Theorem. Therefore, the spectral sequence provides a way to compute these constraints. Detailed explanation of the above statements and examples is presented in the section "Application to HOLSM Theorems".

**Discussion**

In this work, we systematically study real-space construction of TCSs. Starting from building blocks made of SPT states protected by the little symmetry group, we construct a TCS by examining the no-open-edge conditions, the bubbling equivalences, and solving the group-extension problems. These steps form a framework to compute TCS classification. In particular, for bosonic TCS, we prove that, for any symmetry group, this framework gives exactly the same results as the group-cohomology formula in ref. [39]. For the simple cases of $G = SG \times G_0$, the computation is greatly simplified: the classification of topological crystals with building-block dimension $d_b = p$ is given by $E_{p,2}^p = E_{p,\infty}^p$ in Eq. (10).

One advantage of the topological-crystal approach is that it allows us to consider topological crystals with different building-block dimensions separately. In particular, it allows us to consider a more physical classification of crystalline SPTs, which ignores 0D building blocks. The reason for considering this is that when considering the classification of topological states, we usually identify states that can be smoothly deformed to each other without breaking the symmetries. Included in these smooth deformations are insertion and removal of local nondegenerate degrees of freedom, which in general can carry arbitrary 1D linear representations of the local symmetry group. These degrees of freedom are precisely the content of 0D building blocks. In

addition, after removing 0-SPTs, all TCSs in our classification have gapless edges. Therefore, the classification ignoring these 0D building blocks is the more physical one to consider, comparing with the full-group-cohomology classification $H^{d+1}[G, \mathrm{U}(1)_{PT}]$[39].

To demonstrate our framework, we develop an automated code to compute bosonic TCS protected by the direct product of typical onsite symmetries and any of the 2D and 3D crystalline symmetry groups. The results are listed in Supplementary Tables I and II.

Our framework also applies to interacting fermions. Indeed, we give several examples to demonstrate this application in "Methods". The general computation, however, is much more elusive than its bosonic counterpart, due to the complex structure of the space of fixed-point wave functions of onsite symmetries, denoted by $\Psi^d(G_\sigma)$. This is further complicated by the fact that the stacking operation of fermionic wave functions do not necessarily commute, as pointed out in the section "Connectors and second-page results". However, the recent progress in the classification of fermionic SPTs protected by onsite symmetries[17–20,23,25,27] should allow such computation to be carried out. We will leave this to future works.

In the section "Application to HOLSM Theorems", we point out that our framework can also be used to study generalized HOLSM Theorems, especially those enforcing nontrivial SPT states. It will be interesting to apply it to look for new SPT-enforcing HOLSM Theorems in more general symmetry groups that mix crystalline and onsite symmetries. We will also leave this to future works.

*Note added to proof:* As this work was being finalized for posting on the arXiv, refs. [58–60] appeared, which contains some related results.

**Methods**

**No-open-edge conditions on the first page**. Here, we review details of no-open-edge conditions on the first page.

As asserted in the section "No-open-edge condition", the anomaly on a $(p-1)$ cell $\tau$ is computed by directly summing the building blocks on bordering $p$ cells, as in Eq. (6). Several remarks are in order to explain this equation: fFirst, as each $p$ cell has its own local onsite symmetry group $G_{\sigma_i}$, and as the "trivialness" or "nontrivialness" of SPT is only well-defined with respect to some symmetry group, we have to make clear what is the symmetry for the direct sum of the $p$ blocks. The answer is $G_\tau$, the local onsite symmetry group of $\tau$, which is the shared edge of $\sigma_{1,\ldots,n}$. Physically, $\tau$ is a high-symmetry line, and hence has higher symmetry than the bordering high-symmetry planes: $G_{\sigma_i} \subset G_\tau$. Suppose for a pair of $\tau$ and $\sigma$ satisfying $\langle \tau | \partial \sigma \rangle \neq 0$, there is $g \in G_\tau$ and $g \notin G_\sigma$, then $g\sigma$ must be another $p$-cell bordering $\tau$, $\langle \tau | \partial g\sigma \rangle = \langle \tau | \partial \sigma \rangle$. (For better understanding, use Fig. 1a, where $m \in G_{\tau_3}$ and $m \notin G_\sigma$, and $\langle \tau_3 | \partial \sigma \rangle = \langle \tau_3 | \partial m\sigma \rangle = 1$.) The direct sum $[\psi_p|_\sigma] \oplus [\psi_p|_{g\sigma}]$ is hence symmetric under not only under $G_\sigma$ and $G_{g\sigma}$, but also under $g$. Second, we remark that in Eq. (6), if $\langle \tau | \partial \sigma \rangle = -1$, it means that we need to invert the $p$ block at $\sigma$, before stacking it. Last, we also note while $[d_1\psi_\tau^p]$ has the symmetry group of $\tau$, a $(p-1)$ cell, $[d_1\psi_\tau^p]$ is still an SPT (trivial or nontrivial) in $p$ dimensions. Or one can say that $[d_1\psi_\tau^p]$ is a $p$-SPT associated with a $(p-1)$ cell, protected by the symmetries of the $(p-1)$ cell.

We use two examples in bosonic systems to show, respectively, that certain $p$ assemblies in $E_{p,1}^p$ satisfy and do not satisfy the no-open-edge condition. The space group is that of a 3D orthogonal lattice having one mirror plane mapping $x$ to $-x$ plotted in Fig. 2a, and the onsite symmetry group is a unitary $Z_4$ symmetry. We differentiate the cases where (i) the $Z_4$ generator $g_0$ commutes with mirror symmetry $m$: $g_0 m = m g_0$, and (ii) they do not commute $g_0 m = m g_0^{-1}$. Consider an assembly $\psi_2$ generated from decorating $\sigma$ with a 2-block, which is the generator of $\Phi^2(G_{\sigma_1}) = Z_4$, $[\alpha]$. The decorations on the other orbits of $Y_2/G$ are set to be vacuum, and within a unit cell, the only decorated 2-cells are $\sigma$ and $m\sigma$. For case (i), the mirror operation does not act on the local degrees of freedom, $m \cdot [\alpha] = [\alpha]$; but for case (ii), we have $m \cdot [\alpha] = -[\alpha]$. We check the assembly against the no-open-edge condition at $\tau$ between $\sigma$ and $m\sigma$. Following Eq. (6), in case (i) there is

$$[d_1\psi_2|_\tau] = [\alpha] + [\alpha] = 2[\alpha], \tag{12}$$

where we have used $\langle \tau | \partial \sigma \rangle = \langle \tau | \partial g\sigma \rangle$. Since $2[\alpha] \neq 0$, $[\psi_2]$ in case (i) fails the

**Fig. 2 Illustration of no-open-edge condition and bubbling equivalence. a** One symmetric assembly in $E_{2,1}^2$ for space group $Pm$ in one unit cell. The figure shows the cross section of a unit cell that is perpendicular to the $z$ direction. The arrows show the orientation of 2- and 1-cells. This assembly fails the no-open-edge condition. **b** The evolution of assembly in $\ker d_1$ for space group $Pm$ in a fermionic system. The 2-cells are decorated with Chern insulators, the Chern numbers of which are indicated. This state is equivalent to an assembly in $E_{3,1}^2$, where two 3-bubbles are decorated to the 3-cells. The outward (inward) thin arrow means that the Chern number on the associated boundary is $+1$ $(-1)$.

no-open-edge condition. In case (ii), we have

$$[d_1 \psi_2|_\tau] = [\alpha] + (-[\alpha]) = 0. \tag{13}$$

We conclude that $[\psi_2]$ in case (ii) satisfies the no-open-edge condition.

**Bubbling equivalence on the first page**. We now derive the bubbling equivalence in Eq. (9). To show the equivalence relations between assemblies $E_{p,1}^p$ induced by bubbles in $E_{p+1,1}^p$, we need to relate $[\tilde{\psi}_{p+1}]$ to an element in $E_{p,1}^p$. We start with enlarging the bubbles, so that a bubble inside $\nu \in Y_{p+1}$ touches the boundary of $\nu$. The $p$-SPT at the surface of the $(p+1)$ bubble $\nu$ then automatically attaches to all $\sigma \in Y_p$ at the boundary of $\nu$. At the same time, we notice that any given $\sigma$ is the boundary of two or multiple $\nu \in Y_{p+1}$, so that the state induced at $\sigma$ comes from all bordering $\nu \in Y_{p+1}$, which is then computed by Eq. (9).

It is important to realize that although $[\tilde{\psi}_{p+1}|_\nu] \in \Phi^p(G_\nu)$ is a $p$-SPT protected by $G_\nu$, their sum is a $p$-SPT under a larger group $G_\sigma \supset G_\nu$ (after the orientation alignment is resolved by the coefficients $\langle \sigma | \partial \nu \rangle$ in the summation). Therefore, we identify $[\tilde{d}_1 \tilde{\psi}_{p+1}|_\sigma]$ as a $p$-SPT protected by $G_\sigma$, i.e., $[\tilde{d}_1 \tilde{\psi}_{p+1}|_\sigma] \in \Phi^p(G_\sigma) \subset E_{p,1}^p$. Eq. (9) maps $(p+1)$ bubbles to $p$ blocks, establishing a linear map from $E_{p+1,1}^p$ to $E_{p,1}^p$, and since all elements in $E_{p+1,1}^p$ are topologically trivial, the image of the mapping is also trivial. These trivial states in $\mathrm{img}\tilde{d}_1$ give the equivalence relations in $E_{p,1}^p$: $[\psi_p] \sim [\psi_p']$ if and only if $[\psi_p] - [\psi_p'] \in \mathrm{img}\tilde{d}$.

We use the example in Fig. 2b to illustrate the bubble equivalence. It is a fermionic system with charge conservation symmetry, and the lattice is 3D orthogonal with a mirror plane sending $x$ to $-x$. We consider a certain $[\psi_2]$, where some 2-cells are decorated with Chern insulators, the Chern numbers of which are shown in Fig. 2b. It can be easily checked that the assembly in Fig. 2b satisfies the no-open-edge condition. But in Fig. 2b, we show that the state in $E_{2,1}^2$ is actually equivalent to a state made from 3-bubbles only, that is, a state in $\mathrm{img}\tilde{d}_1$. There are two 3-bubbles in one unit cell. The boundary of the left bubble has Chern number $+1$, and that of the right boundary has Chern number $-1$. Therefore this assembly is topologically trivial.

**Connectors and second-page results**. In this section, we explain how to solve the subtleties outlined in the section "Further considerations". A key step in the computations is to determine the concrete content of the connectors decorated on the $(d_b - 1)$ cells, which connect the SPT states on neighboring $d_b$-cells. Using these connectors, we can compute the second-page no-open-edge conditions and bubbling equivalences, which together give the third-page result of the classification. The connector also allows us to solve the group-extension problem arised in the process of combining classifications of TCSs with different $d_b$.

*Contents of the connectors*: We begin by reviewing the content of connectors. Consider a $p$-block TCS $[\psi] \in E_{p,2}^p$, whose building blocks are $p$-dimensional SPT states decorated on the $p$ cell. The connectors on the $(p-1)$ cells are then constrained by these building blocks through the bulk-boundary relation. Previously, we studied the no-open-edge condition in the section "No-open-edge condition," which ensures the existence of gapped symmetric connectors. However, to further determine the concrete form of these connectors, we need not consider only the SPT phases $[\psi_\sigma]$ decorated on $\sigma \in Y_p$, but also the wave functions representing these phases. Just like $[\psi]$ is a collection of local SPT phases, $\psi$ is a collection of local SPT wave functions on all cells: the local SPT phase on $\sigma \in Y_p$ is denoted by $\psi_\sigma \in \Psi^p(G_\sigma)$, where $\Psi^d(G)$ denotes the collection of $d$-dimensional $G$-symmetric wave functions, as reviewed in Sec. I of Supplementary Information. To form a symmetric wave function, we require that the local wave functions decorated to symmetry-related cells satisfy the following symmetry condition, similar to the one

in Eq. (4):

$$\psi_{g\sigma} = g \cdot \psi_\sigma. \tag{14}$$

As $\psi$ is made of $p$-dimensional building blocks, the decorations $\psi_\sigma = 0$ on cells with dimensionality higher than $p$. However, the connectors, which are decorations on cells in dimensions lower than $p$, are in general not vanishing. Collectively, we denote the decorations on $d$ cells by $\psi_d$: $\psi_p$ is the building block on $p$ cells, $\psi_{p-1}$ is the connector on $(p-1)$ cells, etc.

Now consider a $(p-1)$ cell $\tau \in Y_{p-1}$. The connector decorated to $\tau$, which we denote by $\psi|_\tau$, satisfies the following bulk-boundary relation:

$$d(\psi|_\tau) = \boxplus_{\sigma \in Y_p} \langle \tau | \partial \sigma \rangle \psi|_\sigma, \tag{15}$$

where $\psi_1 \boxplus \psi_2$ denotes the wave function obtained by stacking the two wave functions $\psi_1$ and $\psi_2$. As reviewed in Sec. I of Supplementary Information, for bosonic SPT states whose wave functions are represented by cochains, this stacking is just the normal addition between cochains. However, for fermionic SPT states, this stacking operation is not commutative, $\psi_1 \boxplus \psi_2 \neq \psi_2 \boxplus \psi_1$, because the statistical signs are associated with reordering of fermionic operators. Because of this subtlety, to unambiguously define the stacking in Eq. (15), one must choose an ordering between the neighboring cells of $\tau$, and such ordering should be compatible with the crystal symmetries.

For simplicity, we introduce an operator $\partial$ to denote the operation on the right-hand side of Eq. (15): $\partial$ transforms $\psi$ to an anomaly pattern $\partial\psi$, whose components on each cell are given by

$$(\partial\psi)|_\tau = \boxplus_{\sigma \in Y_p} \langle \tau | \partial \sigma \rangle \psi|_\sigma. \tag{16}$$

Intuitively, the operator $\partial$ transfers the wave functions on $p$ cells to their boundary $(p-1)$ cells, where they are interpreted as boundary anomalies. Using this operator, the relation in Eq. (15) is simplied as

$$d(\psi|_\tau) = (\partial\psi)|_\tau. \tag{17}$$

Since such relation exists on every $(p-1)$ cell, it gives a relation between the $p$ blocks $\psi_p$ and the $(p-1)$ connectors $\psi_{p-1}$:

$$d\psi_{p-1} = \partial\psi_p. \tag{18}$$

For bosonic SPT states, the detailed formula for computing the $\partial$ operator can be found in Sec. III of Supplementary Information.

Comparing the right-hand side of Eq. (15) to Eq. (6), it is easy to check that the SPT phase of $(\partial\psi)|_\tau$ is precisely $(\partial\psi)|_\tau \sim d_1[\psi]|_\tau$. Hence, for a second-page TCS $[\psi]$ in $E_{p,2}^p$, the no-open-edge condition $d_1[\psi] = 0$ ensures that Eq. (17) has solutions for $\psi|_\tau$, representing possible choices of a connector bridging $p$ cells bordering $\tau$. In the rest of this section, we shall use solutions of this equation to address the problems raised in the section "Further considerations" and obtain a complete classification of TCSs.

*a. Bosonic example*. We use a simple bosonic example to demonstrate the process of determining the wave functions of connectors. As shown in Sec. V of Supplementary Information, such examples can only be nontrivial when the symmetry group $G$ is not a direct product of $SG$ and $G_0$. In fact, this example involves a magnetic translation symmetry group. This example is adapted from the result of ref. [53]. The connection to ref. [53] will be revealed in the section "Application to HOLSM Theorems".

In this example, we consider 2D TCSs protected by the symmetry group $G = G^M \times \mathbb{Z}_2^T$, where $\mathbb{Z}_2^T$ is the usual (antiunitary) onsite time-reversal symmetry, and $G^M$ is a 2D magnetic translation symmetry group. $G^M$ has three generators $t_x$, $t_y$, $x$, representing two translation symmetries and one onsite unitary $\mathbb{Z}_2$ symmetry, respectively. Both $t_x$ and $t_y$ commutes with $x$. However, $t_x$ and $t_y$ do not commute, and instead satisfy

$$t_x t_y t_x^{-1} t_y^{-1} = x. \tag{19}$$

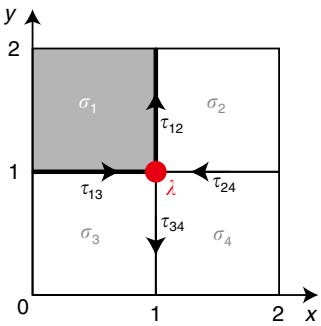

**Fig. 3 The cell decomposition for the magnetic translation symmetry group.** The $x$ and $y$ axes are the two translation axes. $\sigma_i$ label 2-cells, and $\tau_{ij}$ labels the 1-cell between two 2-cells $\sigma_i$ and $\sigma_j$. The red dot labeled by $\lambda$ is a 0-cell.

In this case, the onsite symmetry group $G_0 = \mathbb{Z}_2^x \times \mathbb{Z}_2^T$, where $\mathbb{Z}_2^x$ denotes the $\mathbb{Z}_2$ group generated by $x$. The space group is the quotient group $SG = G/G_0 = \mathbb{Z}^2$, generated by the two translation operations. SZD: $SG$ or $\overline{SG}$? However, $G$ is not a direct product of $G_0$ and $SG$, due to the nontrivial commutation relation in Eq. (19).

We first decompose the 2D plane $\mathbb{R}^2$ into the $G$-complex $Y$ as outlined in the section "Cell decomposition and chain complex". Since $\overline{SG} = \mathbb{Z}^2$ is the simplest wallpaper group, the result of the decomposition is simply a generic oblique lattice, as shown in Fig. 3. There are no point-group symmetries anywhere, and the local symmetry group of each cell is just $G_\sigma = G_0$. Therefore, the SPT building blocks are obtained by attaching SPT states in $\Phi^p(G_0) = H^{p+1}[G_0, U(1)_T]$ to $p$ cells.

In this example, we consider a particular 2-block assembly $[\psi]$ in $E_{2,1}^2$: on each 2-cell, we decorate an SPT state represented by the following cocycle $[\alpha] \in \Phi^2(G_0) = H^3[G_0, U(1)_T]$, represented by the following 3-cocycle:

$$\alpha(g_0, g_1, g_2, g_3) = [n_X(g_0) - n_X(g_1)]\beta(g_1, g_2, g_3)\pi, \tag{20}$$

where the function $\beta$ is defined as

$$\beta(g_0, g_1, g_2) = [n_T(g_1) - n_T(g_0)][n_T(g_2) - n_T(g_1)]. \tag{21}$$

Here, the $\mathbb{Z}_2$ variables $n_x$ and $n_T$ are obtained by writing the elements of $G_0$ as the following canonical form:

$$g = x^{n_x(g)} T^{n_T(g)}. \tag{22}$$

This cocycle represents a nontrivial 2D SPT state protected by both $x$ and $T$ symmetries. The 2-blocks of $[\psi]$ are given by $[\psi]|_\sigma = [\alpha]$.

It is straightforward to check that this element satisfies the cocycle equation on the first page, $d_1[\psi] = 0$, and remains a valid second-page SPT state in $E_{2,2}^2$. To see this, we notice that $[\psi]$ decorates the same SPT state on every 2-cell. Therefore, on each 1-cell, which borders two 2-cells, there are two counterpropagating anomalous edge modes, and they cancel each other. Hence, this decoration $[\psi]$ can be gapped out on 1-cell.

However, gapping out this 1-cell requires nontrivial connectors. In order to compute the connectors, we choose a wave function of 2-blocks $\psi_2$ representing $[\psi]$. As reviewed in Sec. I of Supplementary Information, on each 2-cell, the wave function is a $G$-valued $G_0$-invariant 3-cocycle $\tilde\alpha$. Without losing generality, we choose the following 3-cocycle $\tilde\alpha$:

$$\tilde\alpha(g_0, g_1, g_2, g_3) = [n_x(g_0) - n_x(g_1)]\beta(g_1, g_2, g_3). \tag{23}$$

This equation looks similar to Eq. (20), but the group elements $g_i$ take values in $G$ instead of $G_0$, and $\tilde n_X$ is extracted by writing the group elements in the following canonical form:

$$g = t_x^{n_{tx}(g)} t_y^{n_{ty}(g)} x^{n_x(g)} T^{n_T(g)}. \tag{24}$$

Here, we emphasize that the $\tilde\alpha$ given in Eq. (23) is not invariant under $G$ actions (it is impossible to find a $G$-invariant $\tilde\alpha$ representing this SPT phase, due to the nontrivial structure of $G$.). In fact, using the commutation relation in Eq. (19), one can show that

$$\tilde\alpha(t_y g_0, t_y g_1, t_y g_2, t_y g_3) = [n_x(g_0) + n_{tx}(g_0) - n_x(g_1) - n_{tx}(g_1)]\beta(g_1, g_2, g_3), \tag{25}$$

which is different from Eq. (23).

If we choose to decorate $\sigma_1$ with $\tilde\alpha$ and let $\psi|_{\sigma_1} = \tilde\alpha$, the symmetry constraint will fix the decoration on other 2-cells. In particular, the decoration on $\sigma_3$, which is related to $\sigma_1$ by the action of $t_y^{-1}$, is given by $\psi|_{\sigma_3} = t_y^{-1} \cdot \tilde\alpha$. Using the explicit form of symmetry actions on cochains given in Sec. II of Supplementary Information,

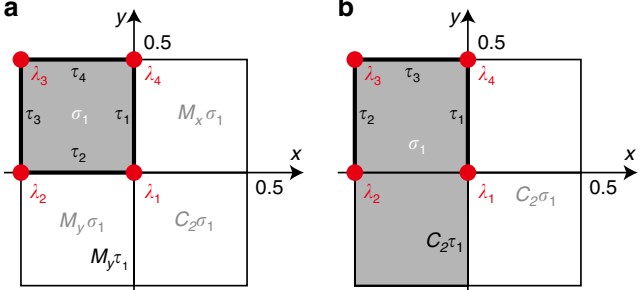

**Fig. 4 Cell decomposition for two wallpaper groups. a** Cell decomposition for the p2mm wallpaper group. $M_x$, $M_y$ and $C_2$ denote the two mirror reflections and the two-fold rotation. **b** Cell decomposition for the p2 group. $C_2$ denotes the two-fold rotation.

we get

$$\psi|_{\sigma_3}(g_0, g_1, g_2, g_3) = \tilde\alpha(t_y g_0, t_y g_1, t_y g_2, t_y g_3). \tag{26}$$

Using the result of Eq. (25), we see that the decorations on the two cells are actually different: $\psi|_{\sigma_1} \neq \psi|_{\sigma_3}$. In fact, the two decorations still belong to the same cohomology class in $\Phi^2(G_0) = H^3[G_0, U(1)_T]$. However, to be compatible with the magnetic translation symmetry, different cochains (of the same cohomology class) have to be decorated to different 2-cells. The difference between $\psi|_{\sigma_1}$ and $\psi|_{\sigma_3}$ then implies that one must decorate a nonvanishing 2-cochain to its boundary, $\tau_{13}$. Using the explicit form of the cochains, one can derive the explicit form of the cocycle equation $d\psi|_{\tau_{13}} = \psi|_{\sigma_3} - \psi|_{\sigma_1}$:

$$d\psi|_{\tau_{13}}(g_0, g_1, g_2, g_3) = [n_{tx}(g_0) - n_{tx}(g_1)]\beta(g_1, g_2, g_3). \tag{27}$$

We now need to choose an arbitrary solution of this equation. It is easy to check that the following 2-cocycle is a choice:

$$\psi|_{\tau_{13}}(g_0, g_1, g_2) = n_{tx}(g_0)\beta(g_0, g_1, g_2). \tag{28}$$

On the other hand, on the 1-cell along the $x$ direction, we are allowed to simply choose $\psi|_{\tau_{12}} = 0$, because the cocycle $\tilde\alpha$ is invariant under the action of $T_x$ and therefore $\psi|_{\sigma_2} = \psi|_{\sigma_1}$. Hence, we have to use nontrivial connectors on some 1-cell, because it is impossible to construct a wave function of $[\alpha]$ that is symmetric under all operations in $SG$.

*b. Free-fermion examples.* Next, we present an example of determining nontrivial connectors in a 2D topological crystaline superconductor. We consider that the 2D system has a wallpaper group $p2mm$, the generators of which are $\hat M_x$ and $\hat M_y$, a time-reversal symmetry, $\hat T$, and a particle–hole symmetry, $\hat P$. The algebra relations of these generators are given by $\hat T^2 = -1$, $\hat P^2 = 1$, $\hat M_x^2 = -1$, $M_y^2 = 1$, $[\hat T, \hat P] = 0$, $[\hat T, \hat M_x] = 0$, $\{\hat T, \hat M_y\} = 0$, $[\hat P, \hat M_x] = 0$, and $[\hat P, \hat M_y] = 0$. Such relations can be realized in a superconductor with significant spin–orbit coupling and an order parameter projectively representing the $\hat M_y$ symmetry. To proceed, we represent these operators as $\hat T = is_2 K$, $\hat P = \mu_1 K$, $\hat M_x = i\mu_3 s_3$, $\hat M_y = \mu_3 s_2$, where $s_{1,2,3}$ are Pauli matrices representing the spin degree, and $\mu_{1,2,3}$ are Pauli matrices representing the "orbital" degree. (The meaning of "orbital" is twisted in BdG Hamiltonian.) The complex structure $Y$ of $p2mm$ is illustrated in Fig. 4a, where the only 2-cell in the $G$ orbits $Y_2/G$ is $\sigma_1$, the four 1-cell in $Y_1/G$ are $\tau_{i=1,2,3,4}$, and the four 0-cell in $Y_0/G$ are $\lambda_{i=1,2,3,4}$.

Now we decorate the 2-cell $\sigma_1$ with the following BdG Hamiltonian:

$$\hat H^{(L)}(\mathbf{k}) = k_x \mu_1 s_3 + k_y \mu_2 s_0 + \left(M - k_x^2 - k_y^2\right)\mu_3 s_0, \tag{29}$$

where $M > 0$. Due to the $\hat M_y$ symmetry, the Hamiltonian decorated on the 2-cell $\sigma_3$ should be $\hat M_y \hat H^{(L)}(k_x, -k_y)\hat M_y^{-1}$, which is identical with Eq. (29). Therefore, we extend the Hamiltonian on $\sigma_1$ (Eq. (29)) to $\sigma_3$. Apparently, there is no boundary state on $\tau_2$. Due to the symmetry $\hat M_x$, the Hamiltonian on $M_x\sigma_1$ and $C_2\sigma_1$ can be derived as

$$\hat H^{(R)}(\mathbf{k}) = \hat M_x \hat H^{(L)}(-k_x, k_y)\hat M_x^{-1} = k_x \mu_1 s_3 - k_y \mu_2 s_0 + \left(M - k_x^2 - k_y^2\right)\mu_3 s_0, \tag{30}$$

which is different from Eq. (29). In order to determine the boundary state between $H^{(L)}$ and $H^{(R)}$, we consider to insert an infinite barrier potential on the edges $\tau_1$ and $M_y\tau_1$ and the vertex $\lambda_1$. We assume that the boundary Hamiltonian on $\tau_1$ is $\hat H^{(\tau_1)} = \mu_0 s_3 k_y$, where the upper block ($\mu_3 = 1$) is the boundary state of $H^{(L)}$ and the lower block ($\mu_3 = -1$) is the boundary state of $H^{(R)}$. The time-reversal symmetry acts locally on the two boundaries, and hence can be represented by $\hat T^{(b)} = is_2 K$. The particle–hole symmetry also acts locally on the two boundaries,

and hence can be represented by $\hat{P}^{(b)} = K$. The mirror symmetry $M_x$ interchanges the two blocks and hence must be proportional to $i\mu_1$ or $i\mu_2$. $M_x$ need to commute with $H^{(\tau_1)}$, $\hat{T}^{(b)}$, $\hat{P}^{(b)}$ and square to $-1$. We find that the only choice is $\hat{M}_x = i\mu_2$. We then soften the barrier to introduce coupling between the two blocks. The only symmetry-allowed mass term that gaps $\hat{H}^{(\tau_1)}$ is $\mu_2 s_1$. Hence we model the gapped state on $\tau_1$ as

$$\hat{H}^{(\tau_1)}(k_y) = \mu_0 s_3 k_y + m\mu_2 s_1. \tag{31}$$

Now we study how $\hat{H}^{(\tau_1)}$ transforms under the $M_y$ operation. Using the constraints $\{\hat{T}^{(b)}, \hat{M}_y^{(b)}\} = 0$, $[\hat{P}^{(b)}, \hat{M}_y^{(b)}] = 0$, $\{\hat{M}_x^{(b)}, \hat{M}_y^{(b)}\} = 0$, $\hat{M}_y^{(b)2} = 1$, $\hat{M}_y^{(b)}$ can be chosen as $\mu_{1,3}s_{1,3}$. Since $M_y$ does not interchange the two blocks, we only consider the two options $\hat{M}_y^{(b)\prime} = \mu_3 s_1$, $\hat{M}_y^{(b)\prime\prime} = \mu_3 s_3$. Correspondingly, the boundary Hamiltonian on $M_y\tau_1$ under the two $M_y$ representations is given by

$$\hat{H}^{(M_y\tau_1)\prime}(k_y) = \hat{M}_y^{(b)\prime}\hat{H}^{(\tau_1)}(-k_y)\hat{M}_y^{(b)\prime-1} = \mu_0 s_3 k_y - m\mu_2 s_1, \tag{32}$$

$$\hat{H}^{(M_y\tau_1)\prime\prime}(k_y) = \hat{M}_y^{(b)\prime\prime}\hat{H}^{(\tau_1)}(-k_y)\hat{M}_y^{(b)\prime\prime-1} = -\mu_0 s_3 k_y + m\mu_2 s_1, \tag{33}$$

respectively. Therefore, either the kinetic term or the mass term will be flipped under $M_y$, leading to a gapless domain wall at $\lambda_1$. This serves as a nontrivial connector on the 1-cell in the vertical direction in Fig. 4a.

In the end, we consider an example of $p + ip$ topological superconductor with nontrivial connector on the 1-cell. We consider a $p + ip$ superconductor in the wallpaper group $p2$. The cell decomposition of $p2$ is shown in Fig. 4b, where the only 2-cell in the $G$ orbits $Y_2/G$ is $\sigma_1$, the three 1-cell in $Y_1/G$ are $\tau_{i=1,2,3}$, and the four 0-cell in $Y_0/G$ are $\lambda_{i=1,2,3,4}$. We assume the Hamiltonian on $\sigma_1$ as

$$\hat{H}^{(\sigma_1)}(\mathbf{k}) = k_x s_1 + k_y s_2 + (M - k_x^2 - k_y^2)s_3, \tag{34}$$

where $M > 0$ and the particle–hole symmetry is represented as $\hat{P} = s_x K$. The Hamiltonian on $C_2\sigma_1$ can be generated by acting the $C_2$ operation on the above Hamiltonian. We consider the $C_2$ operator $\hat{C}_2 = s_0$, it that commutes with $\hat{P}$ and squares to 1. Thus the Hamiltonian on $C_2\sigma_1$ is

$$\hat{H}^{(C_2\sigma_1)}(\mathbf{k}) = \hat{C}_2\hat{H}^{(\sigma_1)}(-\mathbf{k})\hat{C}_2^{-1} = -k_x s_1 - k_y s_2 + (M - k_x^2 - k_y^2)s_3. \tag{35}$$

In order to study the boundary state on $\tau_1$ and $C_2\tau_1$, we consider to insert an infinite barrier potential on $\tau_1$, $C_2\tau_1$, and $\lambda_1$. Since the $H^{(\sigma_1)}(\mathbf{k})$ and $H^{(C_2\sigma_1)}(\mathbf{k})$ have the same chirality (rotation does not change chirality), the Majorana chiral modes from the two sides must move in opposite directions. We assume the boundary Hamiltonian on $\tau_1$ as $\hat{H}^{(\tau_1)} = k_y s_3$, where the $s_3 = 1$ state comes from the boundary of $\sigma_1$ and the $s_3 = -1$ state comes from boundary of $C_2\sigma_1$. Since the particle–hole symmetry acts locally on the two boundaries, we choose its representation as $\hat{P}^{(b)} = K$. Now, we soften the barrier to introduce coupling between the two Majorana modes. The only symmetry-allowed mass term is $s_2$. Hence, we model the gapped state on $\tau_1$ as

$$\hat{H}^{(\tau_1)}(k_y) = s_3 k_y + m s_2. \tag{36}$$

Now we study how $\hat{H}^{(\tau_1)}$ transforms under the $C_2$ rotation. Since $C_2$ interchanges the two boundaries and commutes with $\hat{P}^{(b)}$, its representation must be off-diagonal and real, i.e., $\hat{C}_2^{(b)} = s_1$. The boundary Hamiltonian on $C_2\tau_1$ is hence obtained as

$$\hat{H}^{(C_2\tau_1)}(k_y) = \hat{C}_2^{(b)}\hat{H}^{(\tau_1)}(-k_y)\hat{C}_2^{(b)-1} = s_3 k_y - m s_2. \tag{37}$$

The mass is flipped under the $C_2$ rotation, leading to a gapless domain wall at $\lambda_1$. This serves as a nontrivial connector on the 1-cell.

*Second-page no-open-edge conditions*: For a second-page TCS $[\psi] \in E_{p,2}^p$, the second-page no-open-edge condition demands that all $(p-2)$ cells can be filled with gapped symmetric connectors. The connectors decorated on $(p-2)$ cells, collectively denoted by $\psi_{p-2}$, must satisfy the bulk-boundary relation similar to (18)

$$d\psi_{p-2} = \partial\psi_{p-1}. \tag{38}$$

Hence, the existence of such connectors is determined by the condition that the r.h.s. of Eq. (38) belongs to the trivial SPT phase on each cell, $\partial\psi_{p-1} \sim 0$.

We introduce a linear map $d_{p,2}^p : E_{p,2}^p \to E_{p-2,2}^{p-1}$ to represent this no-open-edge condition. As in "Results", $d_{p,2}^p$ will be abbreviated to $d_2$ if the domain of the map is clear from the context. For each element $[\psi]$ in $E_{p,2}^q$, we choose a particular wave function $\psi_p$ for the building blocks. Then, we choose an arbitrary solution $\psi_{p-1}$ of Eq. (18). The image of $d_2$ map is then defined as

$$d_2[\psi] = [\partial\psi_{p-1}], \quad d\psi_{p-1} = \partial\psi_p. \tag{39}$$

Several remarks are in order: first, the domain of the $d_2$ maps are the $E_2$ modules, because $\psi_{p-1}$ only exists if $[\psi]$ belongs to the $E_2$ module, where the cocycle condition $d_1[\psi] = 0$ guarantees the existence of $\psi_{p-1}$. Second, we explain

why the images of $d_2$ maps are the $E_2$ modules. The meaning of this assertion is twofold: On one hand, $d_2[\psi] = [\partial\psi_{p-1}]$ satisfies the cocycle condition $d_1 d_2[\psi]$ because $\partial^2 = 0$. Therefore, it indeed belongs to $E_2$. On the other hand, the equivalence relation of $d\psi_{p-2} = \partial\psi_{p-1} \sim 0$ should be understood as the one in $E_{p-2,2}^{p-1}$: Instead of requiring the obstruction $[(\partial\psi_{p-1})|_\sigma]$ to vanish on every $(p-2)$-cell $\sigma$, we only require that $\partial\psi_{p-1}$ can be trivialized by a bubbling process $\tilde{\psi} \in E_{p-1,1}^{q+1}$: $\partial\psi_{p-1} \sim d_1[\tilde{\psi}] = \partial\tilde{\psi}_p$. This is because when extending $\psi$ to $(p-1)$-cells, we can choose $\psi'_{p-1} = \psi_{p-1} - \tilde{\psi}_p$ instead of $\psi_{p-1}$ as the connectors. This choice of connectors then satisfies $(\partial\psi'_{p-1})|_\mu \sim 0$ on every $(p-2)$-cell.

In summary, the second-page no-open-edge condition, which tests whether $[\psi]$ can be filled with gapped symmetric connectors on $(p-2)$-cells, is expressed as

$$d_2[\psi] = 0. \tag{40}$$

*a. Bosonic example.* We now use this no-open-edge condition to examine the example introduced in the section "Bosonic example". We will see that the $d_2$ map is nontrivial in this example. We consider the result of $d_2[\psi]$ on the $\mu$ shown in Fig. 3. Recall that using the wave-function realization $\psi$ we chose in Eq. (23), the connectors $\psi_1$ are given as follows: $\psi_1|_{\tau_{12}} = \psi_1|_{\tau_{34}} = 0$; $\psi_1|_{\tau_{13}}$ is given by Eq. (28). Hence, $\psi_1|_{\tau_{24}}$ is constrained to be $t_x^{-1} \cdot \psi_1|_{\tau_{13}}$ and has the following form,

$$\psi_1|_{\tau_{24}}(g_0, g_1, g_2) = [n_{tx}(g_0) + 1]\beta(g_0, g_1, g_2). \tag{41}$$

Using these results of $\psi_1$, we can compute $d_2[\psi]$ using the definition in Eq. (39),

$$d_2[\psi]|_\tau \sim \psi_1|_{\tau_{13}} + \psi_1|_{\tau_{24}} + \psi_1|_{\tau_{12}} + \psi_1|_{\tau_{34}} \sim [\beta]. \tag{42}$$

Here, $\beta$ is exactly the 2-cocycle representing the projective representation of $T^2 = -1$, or a Kramers doublet. Hence, in this case, $d_2 : E_{2,2}^3 \to E_{0,2}^2$ is a nontrivial map that sends $\hat{\alpha}$ to the nontrivial element $[\tilde{\psi}]$ in $E_{0,2}^2$:

$$d_2[\psi] = [\tilde{\psi}]. \tag{43}$$

Here, $[\tilde{\psi}]$ is an anomalous pattern represented by the following decomposition on 0-cells: $[\tilde{\psi}]_\mu = [\beta]$ on all 0-cells in $Y$. This implies that $[\psi]$ is actually not a 2D TCS because it does not satisfy the second-page no-open-edge condition: it has open edges, actually Kramers doublets, on the 0-cells.

*Free-fermion example*: Next, we visit the free-fermion example in the section "Free-fermion examples". In this example, we use the connector in Eq. (31) to gap out the 1-cells in the $y$ direction in Fig. 4. Furthermore, the symmetry condition in Eq. (14) implies that mass terms on different 1-cells are related by symmetries. In particular, using the projected symmetry operation $\hat{M}_y^{(b)} = s_2$, we see that the mass term must change sign under $\hat{M}_y^{(b)}$. Therefore the mass terms on the two cells $\tau_1$ and $\tau_3$, related by $M_y$, must have opposite signs. Such two opposite mass terms would left a symmetry protected gapless point at $y = 0$, i.e., the 0-cell $\lambda_1$. Replacing the mirror symmetries above with mirror symmetries on $x = -1/2$ and $y = 1/2$, and following the same analysis, one can easily show that all the 0-cells in $Y_0/G$, $\lambda_{1,2,3,4}$, are gapless.

*a. Second-page bubbling equivalences.* The second-page bubbling equivalence can be computed using a similar $d_2$ map. Each element $[\psi] \in E_{p+2,2}^{p+1}$ represents such a bubbling process, where SPT bubbles are generated on $(p+2)$-cells. Unlike the bubbling processes studied in the section "Bubble equivalence", it leaves the $(p+1)$-blocks intact, and changes the SPT phases on the $p$-cells.

In order to compute the changes to the $p$-cells, we also need to consider additional bubbles generated on lower-dimensional cells. On a $p'$-cell $\tau \in Y_{p'}$ $(p'+2)$, we can also generate a bubble $\psi|_\tau$, which we will refer to as a $p'$-bubble. Different from the $(p+2)$-bubbles, the lower-dimensional bubble can have a nontrivial filling: $d(\psi|_\tau) \neq 0$, meaning that the process not only changes the wave functions on $\partial\tau$ by $\psi|_\tau$, but also changes the wave function on $\tau$ by $d(\psi|_\tau)$. Such a process is allowed because the bubble and its filling satisfy the bulk-boundary relation reviewed in Sec. I of Supplementary Information, and together form a gapped symmetric state on $\tau$. In this way, a general bubbling process contains not only $p$-bubbles, denoted by $\psi_p$, but also $p'$-bubbles $\psi_{p'}$ for all $p'$. On $p'$-cells, the total changes made by the bubbling include the $(p'+1)$-bubbles and the filling of the $p'$-bubbles:

$$\Delta\psi_{p'} = \partial\psi_{p'+1} \boxplus d\psi_{p'}. \tag{44}$$

We now compute the changes to $p$-cells for a bubbling process $[\psi] \in E_{p+2,2}^{p+1}$. First, we choose a wave-function realization $\psi_{p+2}$ of the $(p+2)$-bubbles. Since the bubbling process leaves $(p+1)$-blocks intact, the $(p+1)$-bubble $\psi_{p+1}$ must satisfy

$$\partial\psi_{p+2} \boxplus d\psi_{p+1} = 0. \tag{45}$$

The existence of solutions of this equation is provided by the fact that $[\psi] \in E_{p+2,2}^{p+1}$ satisfies $d_1[\psi] = 0$. We then choose an arbitrary solution of $\psi_{p+1}$, and the SPT phases the process generates is $d\psi_p \boxplus \partial\psi_{p+1} \sim [\partial\psi_{p+1}]$. Therefore, this process can be represented by the following $d_2$ map,

$$d_2[\psi] = [\partial\psi_{p+1}], \quad d\psi_{p+1} = -\partial\psi_{p+2}. \tag{46}$$

Note, that the $d_2$ map defined here is different from the one defined in Eq. (39) by a minus sign in the constraint equation. Hence, we can use a generic definition to

unify the two $d_2$ maps:

$$d^q_{p,2}: E^q_{p,2} \to E^{q-1}_{p-2,2}: [\psi] \mapsto [\partial\psi_{p-1}], \quad d\psi_{p-1} = (-1)^{q-p}\partial\psi_p. \quad (47)$$

We notice that a nontrivial example of $d_2$ bubbling process is provided in the section "Application to HOLSM Theorems".

Taking into account the no-open-edge conditions and bubbling equivalences given by the $d_2$ map, the third-page approximation of the TCS classification is given by the cohomology group of $d_2$:

$$E^q_{p,3} = \frac{\ker d^q_{p,2}}{\operatorname{img} d^{q+1}_{p+2,2}}. \quad (48)$$

**Higher-page results**. This process can be generalized to higher pages.

To consider the $r$th page no-open-edge conditions, we start with an $r$th page assembly $\psi \in E^q_{p,r}$. On previous pages, we have constructed the connectors $\psi_{p-1}, ..., \psi_{p-r+1}$. The no-open-edge condition on the previous page, $d_{r-1}[\psi] = 0$, guarantees that the equation $d\psi_{p-r} = \partial\psi_{p-r+1}$ has solutions. We then pick a solution of $\psi_{p-r}$, and define $d_r[\psi] = [\partial\psi_{p-r}] \in E^{p-r+1}_{p+r,r}$. The no-open-edge condition is given by $d_r[\psi] = 0$.

Similarly, consider an $r$th page bubbling process $[\psi] \in E^{p+r-1}_{p+r,r}$. On previous pages, we have chosen lower-dimensional bubbles $\psi_{p+r-2}, ..., \psi_{p+2}$, such that the process generates nothing on cells with dimensions higher than $p+2$. In order to get a process that also generates nothing on $(p+1)$-cells, we choose $\psi_{p+1}$ satisfying $d\psi_{p+1} \boxplus \partial\psi_{p+2} = 0$. The existence of solutions is provided by $d_{r-1}[\psi] = 0$. The resulting trivial TCS is given by $d_r[\psi]$ defined as $[\partial\psi_{p+1}]$.

Similar to Eq. (47), we write a unified definition for $d_r$:

$$d^q_{p,r}: E^q_{p,r} \to E^{q-r+1}_{p-r,r}: [\psi] \mapsto [\partial\psi_{p-r+1}], \quad d\psi_{p-r+1} = (-1)^{q-p}\partial\psi_{p-r+2}. \quad (49)$$

Therefore, the classification on the next page is given by the cohomology group of $d_r$:

$$E^q_{p,r+1} = \frac{\ker d^q_{p,r}}{\operatorname{img} d^{q+r-1}_{p+r,r}}. \quad (50)$$

Iteratively, this process computes a series of pages $E^q_{p,1}, E^q_{p,2}, ...$, where $E^q_{p,r}$ provides a series of finer and finer approximations to the classification of $p$-block TCSs. Since we are eliminating false and redundant entries on each page, the list of candidate assemblies are getting smaller and smaller, $E^q_{p,1} \supseteq E^q_{p,2} \supseteq \cdots$. In the limit of $r \to \infty$, this series of approximations reveals the true answer of the classification problem, which we denote by $E^q_{p,\infty}$. In particular, $E^q_{p,\infty}$ classify all $p$-block TCSs. In fact, this process only takes a finite number of steps to converge to $E^q_{p,\infty}$, because the $d_r$ map reduces the spatial dimension of the cells by $r$ and necessarily becomes trivial once $r$ exceeds the dimension of $Y$.

**Recombing states with different $d_b$ through group extension**. In our topological-crystal constructions, we first divide $d$-dimensional SPTs into TCSs with different building-block dimensions $d_b = 0, 1, ...d$. We then compute the classification for each $p$-blocks separately. The classification for $p$-block TCSs is then given by $E^p_{p,\infty}$, which is calculated by a series of cohomology-group computations. Next, to obtain the full classification of crystaline SPT states, we need to recombine results of $d_b = 1, 2, ...d$. We also need to include results of $d_b = 0$ if we want to recover all bosonic SPTs in $H^{d+1}[G, U(1)_T]$. However, as briefly mentioned in the section "Further considerations", such recombination may not be a simple direct sum but a nontrivial group extension. In this section, we explain how this group extension is computed in general. Sec. V of Supplementary Information will use this method to prove that, for the simple cases $G = SG \times G_0$, the group extension is always trivial and one can just take a direct sum.

We begin by recalling that a TCS $E^p_{p,\infty}$ are labeled by different building blocks on $p$-cells, but each state $[\psi]$ also contains connectors on all lower-dimensional cells. The lower-dimensional decorations will affect the results of adding (stacking) two SPTs states if the decoration cancels on higher-dimensional cells.

To be more specific, consider an order-$n$ TCS $[\psi] \in E^p_{p,\infty}$, such that $n[\psi] \sim 0$ as in $E^p_{p,\infty}$. This implies that stacking $n$ copies of $[\psi]$ results in a state $[\tilde\psi] = n[\psi]$ which is trivial if viewed as an element in $E^p_{p,\infty}$. In other words, $[\tilde\psi]$ has trivial decorations on all $p$-cells. However, $[\tilde\psi]$ may have nontrivial decorations on lower-dimensional cells, and thus should be viewed as a nontrivial topological crystal with a lower building-block dimension. To compute this, recall that in the topological crystal $[\psi]$, the subleading terms $\psi_{p'}$, representing the decoration on $p'$-cells, are obtained in the spectral-sequence computation. Using these subleading terms, the decorations on lower-dimensional cells in $\tilde\psi = n\psi$ is computed as

$$\tilde\psi_{p-r} = n\psi_{p-r}. \quad (51)$$

One can then look $\tilde\psi_{p-r}$ up in $E^{p-r}_{p-r,\infty}$ to see whether it is nontrivial. The smallest $r$ such that $\tilde\psi_{p-r}$ is nontrivial then indicates $\tilde\psi = n\psi$ is a nontrivial TCS with $(p-r)$-blocks. When this happens, combining $E^p_{p,\infty}$ and $E^{p-r}_{p-r,\infty}$ then becomes a nontrivial group-extension problem instead of a direct sum.

We notice that the answer of whether $n[\psi]$ is a nontrivial SPT state may depend on the choice of the subleading terms of the generator $\psi$ (on the contrary, the computation of $E^p_{p,r}$ does not depend on this choice). However, the final result of the group-extension problem is independent of the choice of the generators. In Sec. V of Supplementary Information, we will show that for the simple cases of bosonic TCSs $G = SG \times G_0$, if we choose to ignore the 0-block TCS, there exists a simple choice of $\psi_{p-1}$ such that the group-extension problem becomes trivial, and a naive direct sum gives the correct classification.

**Examples**. The algorithm to classify TCSs outlined in above sections can be automated for the bosonic case, using the formulation given in SI. For simplicity, here we only consider the case when the total group is a direct product of space (wallpaper) group and a local symmetry group, i.e., $G = SG \times G_0$. In this case, since $d_2$ map is trivial, we only need to take care of the first-page no-open-edge condition and the bubble equivalence. By an automated script, we have enumerate the bosonic TCSs with seveal local symmetry groups in all the wallpaper groups and *all* the space groups. The main results can be found in Supplementary Tables I and II, respectively.

In our results, we find that, although some of the 2D constructions of TCSs are equivalent to decoupled straight lines or straight planes, some of them, however, are beyond such simple layer constructions. Here, we give a bosonic example of a TCS beyond layer constructions, which has a geometric structure similar to an example studied in ref. [36]. We consider the space group $P\bar{4}n2$ with the local symmetry group $Z_2$. The 2-cells and 1-cells are shown in Fig. 5. The details of the 2-cells and 1-cells are given in Sec. VI of Supplementary Information. According to refs. [3,4,61], $Z_2$ symmetry protects a $\mathbb{Z}_2$ 2D SPT. Thus we can decorate each 2-cell with a such a 2D SPT. As discussed in Sec. V of Supplementary Information, the anomalies of these 2D SPTs can cancel each other on the 1-cells where they meet. Thus, the no-open-edge condition reduces to the constraint that there should be even number of 2D SPT ending at each 1-cell[36]. On the other hand, the bubble equivalence is trivial because on the 2-cells the $\mathbb{Z}_2$ bubble is always canceled by its symmetry partner[36]. Therefore, the classification of bosonic TCS in this case is just given by the allowed decorating configurations. The boundaries of the four

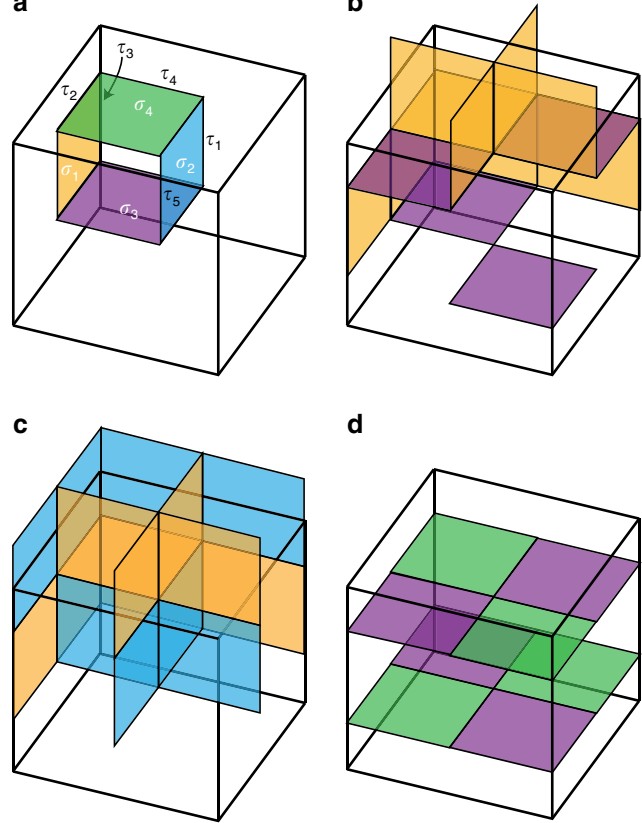

**Fig. 5 A 2D construction beyond layer construction. a** The inequivalent 2-cells and 1-cells of the space group $P\bar{4}n2$. **b–d** The three $\mathbb{Z}_2$ generators for the crystalline states. **b** is beyond layer construction, and (**c, d**) are equivalent with layer constructions. The equivalent 2-cells are colored in the same colors. See Sec. VI of Supplementary Information for the definitions of the 2-cells and 1-cells.

**Fig. 6 Cell decomposition for a 1D system with a mirror-reflection symmetry.** $\sigma_1$ and $\sigma_2$ denotes two 1-cells related by the mirror symmetry $M$, and the dot labeled by $\tau$ is the 0-cell between them.

inequivalent 2-cells are

$$\sum_{g \in SG} \partial g \sigma_1 = \sum_{t \in T} t\tau_2 + t\tau_4, \tag{52}$$

$$\sum_{g \in SG} \partial g \sigma_2 = \sum_{t \in T} t\tau_2 + t\tau_4, \tag{53}$$

$$\sum_{g \in SG} \partial g \sigma_3 = \sum_{t \in T} t\tau_2 + t\tau_4, \tag{54}$$

$$\sum_{g \in SG} \partial g \sigma_4 = \sum_{t \in T} t\tau_2 + t\tau_4, \tag{55}$$

where $T$ represents the translation group. Since all the 2-cells have the same boundary, in order to cancel the anomalies on the 1-cell we only need to decorate two of the four 2-cells. There are three independent decorations: $\sigma_1 + \sigma_3$, $\sigma_1 + \sigma_2$, $\sigma_3 + \sigma_4$. As shown in Fig. 5b–d, $\sigma_1 + \sigma_2$ and $\sigma_3 + \sigma_4$ decompose into horizontal and vertical planar layers, whereas $\sigma_1 + \sigma_3$ does not. In fact, using similar argument in ref. [36], one can show that $\sigma_1 + \sigma_3$ is not equivalent with any layer construction.

Next, we demonstrate the application of our real-space recipe to classify interacting-fermion SPTs by revisiting an example discussed in ref. [62]. In this example, we consider a 1D superconducting system with a time-reversal symmetry $T$ and a mirror-reflection symmetry $M$ that reverses the whole 1D system, and we assume that the fermions transform as spinless fermions under both $T$ and $M$ (i.e. fermions carry linear representations $T^2 = M^2 = +1$). When considering interacting-fermion classification, being a superconducting system means there is no U(1) charge-conservation symmetry. To classify topological superconductors in such a system, we divide the 1D system into two 1-cells $\sigma_{1,2}$ and a 0-cell $\tau$, as shown in Fig. 6. The 0-cell is located at the reflection center, and the two 1-cells are mapped to each other by the reflection symmetry $M$. In 1D, the TCSs we are interested in are given by $E_{1,\infty}^1$, denoting decorating 1D blocks on 1-cells (we are ignoring 0D decorations in $E_{0,\infty}^0$, as we always do in this paper.)

On the first page, $E_{1,1}^1$ is given by Eq. (5). Here, $Y_1/G$ contains only one $G$-orbit containing both $\sigma_{1,2}$. Choosing $\sigma_1$ to represent this orbit, $E_{1,1}^1$ is given by $\Phi^1(G_{\sigma_1})$, which contain 1D SPT states protected by the onsite symmetry group $G_{\sigma_1} = \mathbb{Z}_2^T$. It is well-known that $\Phi^1(\mathbb{Z}_2^T)$, representing the phases 1D topological superconductors with $T^2 = +1$, has a $\mathbb{Z}_8$ classification, where the root state is the 1D Kitaev chain[63,64]. Hence, we have $E_{1,1}^1 = \mathbb{Z}_2$.

Next, we compute the first-page no-open-edge condition. (There is no bubbling equivalence in this case, because there is no 2-cell.) This condition is described by the first-page derivative $d_{1,1}^1 : E_{1,1}^1 \to E_{0,1}^1$. Here, the codomain of this derivative is the space of anomaly patterns on the 0-cell $\tau$: $E_{0,1}^1 = \Phi^1(G_\tau)$. As the reflection center, $G_\tau = \mathbb{Z}_2^T \times \mathbb{Z}_2^M$, where $M$ is a unitary onsite symmetry. Correspondingly, the classification of anomaly patterns is given by $\Phi^1(G_\tau) = \mathbb{Z}_8 \oplus \mathbb{Z}_4$, where the $\mathbb{Z}_8$ and the $\mathbb{Z}_4$ subgroups are generated by the same Kitaev chain and an additional fermionic SPT state with complex-fermion decorations, respectively[65]. The second root state does not play a role in our calculation, so its details will not be discussed here. Representing elements of $E_{1,1}^1$ and $E_{0,1}^1$ by a mod-8 integer $n$ and a pair of a mod-8 integer and a mod-4 integer $(n_1, n_2)$, respectively, the derivative $d_{1,1}^1$ has the form $d_{1,1}^1(n) = (2n, 0)$. The first component of $d_{1,1}^1(n)$ is $2n$, because $\sigma_1$ and $\sigma_2$ should be decorated by the same state denoted by $n$ as the result of the symmetry condition, and their total contribution to the anomaly on $\tau$ is thus $2n$. The second component of $d_{1,1}^1(n)$ can be computed explicitly using the approach in Sec. IC of Supplementary Information, but this result does not affect the classification of $E_{1,\infty}^1$, and hence will not be discussed here. Using the explicit form of $d_{1,1}^1$, we see that the no-open-edge condition $d_{1,1}^1(n) = 0$ demands that $2n = 0 \mod 8$, or $n = 0, 4$. Therefore, the second-page result is $E_{1,2}^1 = \ker d_{1,1}^1 = \mathbb{Z}_2$, generated by $n = 4$, representing decorating four copies of Kitaev chains on $\sigma_{1,2}$. In 1D, $E_{1,2}^1 = E_{1,\infty}^1$ is the final answer. Hence, the topological superconductors in such a 1D system has a $\mathbb{Z}_2$ classificationm, and the root state is the interaction-enabled topological superconductor state studied in ref. [62].

The steps described above can be generalized to study interacting-fermion SPT states with more complex space-group symmetries. Some examples of using similar ideas to study real-space construction of fermionic topological crystalline states can be found in refs. [41,62,66]. We shall leave a fully automated implimentation of our real-space recipes to future works.

**Application to HOLSM theorems.** In this section, we apply our TCS constructions to obtain generalized HOLSM-type theorems. We first review the original HOLSM Theorem and its generalizations to different onsite symmetry groups, and discuss how to understand them using our TCS constructions. We then revisit the first example given in the section "Second-page no-open-edge conditions", and discuss how to reinterpret it as an SPT-enforcing HOLSM Theorem[53,57]. Last, we will give the general relation between TCS constructions and generalized HOLSM Theorems. For simplicity, we first discuss the 2D examples, then generalize our results to 3D.

The original HOLSM Theorem asserts that in a 2D lattice with translation symmetries and spin-rotation symmetries, if there is an odd number of spin-$\frac{1}{2}$ per unit cell, the system cannot have a symmetric gapped unique ground state. This theorem is later generalized to the cases where the spin-rotation symmetry and the spin-$\frac{1}{2}$ objects are replaced by an arbitray onsite symmetry group $G_0$ and a nontrivial projective representation of $G_0$, respectively. In this section, we will refer to the original and these generalizations as the "generalized HOLSM Theorems".

Using our TCS framework, we can view the distribution of projective representations as a nontrivial anomaly pattern $[\tilde\psi]$ in the module $E_{0,\infty}^1$. In our language, the total symmetry group is $G = SG \times G_0$, where the space group $SG = \mathbb{Z}^2$. Hence, the $G$-complex $Y$ we construct is the same as the one shown in Fig. 3, with one 0-cell per unit cell. The local symmetry group on the 0-cell is simply $G_0$. The nontrivial projective representation can be translated to a nontrivial element $[\beta] = H^2[G_0, U(1)_T] = \Phi^1(G_0)$, which is decorated to the 0-cells, as $[\tilde\psi]_\mu = [\beta]$. In this way, the distribution of a nontrivial projective representation is translated to an anomaly pattern $[\tilde\psi]$.

We now argue that, including this $[\tilde\psi]$, every nontrivial element in $E_{0,\infty}^1$ represents an anomaly pattern that cannot be gapped out by a symmetric unique ground state. This is done by reinterpreting the no-open-edge conditions we introduced in the section "Connectors and second-page results". Consider an element $[\tilde\psi]$ in $E_{0,r}^1$ that is trivialized by the $d_r$ map,

$$d_r[\psi] = [\tilde\psi]. \tag{56}$$

In the section "Connectors and second-page results", we interpreted this relation as the fact that, the assembly $[\psi]$ does not satisfy the no-open-edge condition and cannot be realized as a TCS, because assembling it will result in anomaly patterns specified by $[\tilde\psi]$ on the 0-cells. However, if the physical Hilbert space already contains an anomaly pattern $[\tilde\psi]$ on the 0-cells, the assembly $-[\psi]$ can be realized in such physical systems, because the obstruction $d_r(-[\psi]) = -[\psi]$ is now canceled by the background anomaly pattern in the Hilbert space. Therefore, Eq. (56) also implies that the anomaly pattern $-[\psi]$ can be gapped out by the TCS assembly $[\psi]$. In other words, it reveals a UV/IR anomaly matching between the TCS assembly (which can be viewed as the IR limit) and the anomaly pattern (which can be viewed as the UV limit).

A corollary of this reinterpretation is that a nontrivial element in $E_{0,\infty}^1$ cannot be gapped out by any such TCS assembly, and therefore cannot realize a symmetric gapped unique ground state.

We can also revisit the first example in the section "Second-page no-open-edge conditions" using this alternative interpretation. Recall that Eq. (43) indicates that $[\psi]$ does not represent a valid 2D $G$-SPT state: tiling the 2D plane with the SPT phase $[\alpha]$ will leave one gapless Kramers doublet in each unit cell. However, if we start with a model that has one Kramers doublet per unit cell in the original Hilbert space, this will cancel the anomaly of $d_2[\psi]$ and allows the construction of the 2D SPT $[\alpha]$. The construction of the trivial SPT state in such system, however, becomes impossible, because the anomaly would be left uncanceled. In other words, the existence of a nontrivial anomaly pattern in the Hilbert space requires that a gapped unique ground state must be a nontrivial SPT state. This is precisely the theorem proved in ref. [53], which we will refer to as an SPT-enforcing theorem.

In general, we can express these HOLSM-like constraints as the following: In our language, a spatial distribution of projective representations is represented by an anomaly pattern $[\tilde\psi]$ in $E_{0,1}^1$. If $[\tilde\psi]$ is trivialized through $d_1$ by a $[\psi] \in E_{1,1}^1$ as $[\tilde\psi] = d_1[\psi]$, then $[\tilde\psi]$ can be gapped out by the 1-block TCS assembly $[\psi]$. If $[\tilde\psi]$ is a nontrivial element in $E_{0,2}^1$ but is trivialized through $d_2$ by $\hat\alpha \in E_{2,2}^2$ as $[\tilde\psi] = d_2[\psi]$, then $[\tilde\psi]$ can be gapped out by the 2-block TCS assembly $[\psi]$, which must be a strong SPT state (i.e., it is protected solely by $G_0$) because 2-cells in a 2D space must have $G_\sigma = G_0$. Furthermore, if $[\tilde\psi]$ is a nontrivial element in $E_{0,3}^1 = E_{0,\infty}^1$, then it cannot be gapped out by an SPT state, and will give the consequences of the original LSM Theorem. Therefore, the spectral sequence introduced in the section "Connectors and second-page results" provides a way to compute these constraints.

These constraints can be further generalized to 3D. There, an anomaly pattern, which is an element in $E_{0,1}^1$, can be trivialized through $d_1$ by a TCS assembly in $E_{1,1}^1$, through $d_2$ by a TCS assembly in $E_{2,2}^2$, through $d_3$ by a TCS assembly in $E_{3,3}^3$, or cannot be trivialized at all. Among these possiblities, $E_{3,3}^3$ represents strong 3D SPT states.

The UV/IR anomaly matching condition in Eq. (56), and the resulting SPT-enforcing HOLSM Theorems, can be understood by viewing the anomaly pattern as the surface anomaly of a 3D bulk state. We first explain this for the simple cases of the generalized HOLSM Theorem with $G = G_0 \times \mathbb{Z}^2$. Since the projective representation $[\beta]$ is the edge state of a 1D SPT state, the 3D bulk state can be

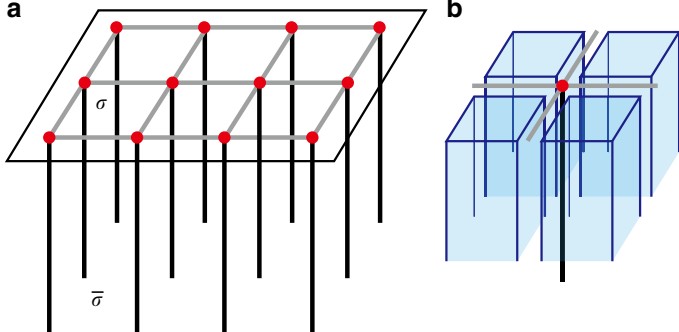

**Fig. 7 Mapping between HOLSM in 2D and TCS in 3D. a** Mapping between a 2D $G$-complex $Y$ and a 3D $G$-complex $\bar{Y}$. $Y$ can be viewed as the boundary of $\bar{Y}$. **b** Illustration of a 3-block bubbling process in $\bar{Y}$. On the second page, such a process trivializes a 1-block assembly in $\bar{Y}$ and generates a 2-block SPT in $Y$.

constructed by decorating one such SPT chain in each 3D unit cell, as shown in Fig. 7a. Here, we argue that the 3D bulk state is closely related to $[\tilde{\psi}]$, and can be constructed mathematical from it. The 3D bulk has the same symmetry $G$ as its surface. Hence, we construct a 3D topological space $\bar{Y}$ compatible with $SG$. $\bar{Y}$ has one 3-cell, two 2-cells and one 1-cell in each unit cell (it has no 0-cells). The aforementioned 3D bulk state is represented by an SPT pattern $[\bar{\psi}] \in \bar{E}^1_{1,\infty}$, constructed by decorating one Haldane chain on each 1-cell. Here, the bar on $\bar{E}^q_{p,r}$ indicates that it is the spectral sequence constructed for $\bar{Y}$ instead of $Y$. It is easy to realize that there is a one-to-one correspondence between $p$-cells in $Y$ and $(p+1)$-cells in $\bar{Y}$, which can be expressed as an isomorphism $Y_p \simeq \bar{Y}_{p+1}$. We denote the corresponding cells in $Y_p$ and $\bar{Y}_{p+1}$ as $\sigma$ and $\bar{\sigma}$, respectively. Using $\bar{Y}$, the SPT pattern $[\bar{\psi}]$ can be expressed as decorating one Haldane chain to each 1-cell in $\bar{Y}$. Next, we notice that, mathematically, the anomaly decorated to each $\sigma \in Y_0$ and the 1D SPT decorated to each $\bar{\sigma} \in \bar{Y}_1$ are represented by the same 2-cocycle $[\beta]$ in $H^2[G_0, U(1)_T]$. Hence, $[\tilde{\psi}]$ and $[\bar{\psi}]$ can be converted to each other by copying the decoration between corresponding cells. Mathematically, this is described by the fact that the isomorphism $Y_p \simeq \bar{Y}_{p+1}$ naturally induces an isomorphism $E^1_{0,\infty} \simeq \bar{E}1_{1,\infty}$: the isomorphsm map, denoted by $L : E^1_{0,\infty} \rightarrow \bar{E}^1_{1,\infty}$, is given by

$$L([\tilde{\psi}])|_{\bar{\sigma}} = [\bar{\psi}]|_{\sigma}. \tag{57}$$

Using this isomorphism, the relation between the anomaly pattern $[\tilde{\psi}]$ and the corresponding bulk state $[\bar{\psi}]$ is then given by $\bar{\psi} = L([\tilde{\psi}])$.

The surface-bulk correspondence illustrated above can be generalized to arbitrary symmetry groups and dimensions. We consider a 2D surface and a 3D bulk with the same symmetry group $G$. Here, the space group $SG = G/G_0$ is a 2D wallpaper group instead of a 3D space group. Similar to the previous example, the cellular decomposition for the 3D bulk can be constructed from the one of the 2D surface using the isomorphism $\bar{Y}_{p+1} \simeq Y_p$. Again, this induces an isomorphism between $E^q_{p,r}$ and $\bar{E}^q_{p+1,r}$ for arbitrary $p$, $q$, and $r$ through the definition in Eq. (57). To understand the physical meaning of this correspondence, we notice that the $p$-cell $\sigma \in Y_p$ is an edge of the $(p+1)$-cell $\bar{\sigma} \in \bar{Y}_{p+1}$. In this way, for $[\tilde{\psi}] \in E^{p+1}_{p,r}$, representing an $r$-th-page anomaly pattern on the surface, $L([\tilde{\psi}])$ represents decorating on $\bar{\sigma}$ the bulk SPT state $[\bar{\psi}]|_{\sigma}$ that corresponds to the anomaly $[\tilde{\psi}]|_{\sigma}$ on its boundary $\sigma$. Hence, $L([\tilde{\psi}]) \in \bar{E}^{p+1}_{p+1,r}$ is the bulk SPT state corresponding to the surface anomaly pattern $\hat{w}$. Similarly, consider a surface TCS $[\psi] \in E^{p+1}_{p,r}$. $L([\psi]) \in \bar{E}^{p+1}_{p+1,r}$ is a bubbling pattern that generates SPT states $[\psi]$ on the edges of cell $\bar{\sigma}$, including $\sigma$ itself. Hence, the 3D bubbling pattern $L([\psi])$ generates the 2D TCS $[\psi]$ on the surface.

In summary, the isomorphism $L$ defined above allows us to express the correspondence between 2D anomaly patterns and 3D TCSs, and between 2D TCSs and 3D bubbling patterns. It also allows us to convert the 2D no-open-edge conditions to 3D bubbling equivalences. Consider a $d_r$ map between a 2D assembly and a 2D anomaly pattern, as in Eq. (56). The $L$-isomorphism maps the r.h.s. to a 3D TCS that can host the anomaly pattern on its 2D surface, and the r.h.s. to a 3D bubbling pattern that generates the 2D assembly on its surface (see Fig. 7b). In this way, the relation (56) also describes a 3D bubbling equivalence,

$$d_r L([\psi]) = L([\tilde{\psi}]). \tag{58}$$

This dimensional-shifting correspondence is consistent with our understanding that the TCS $L([\tilde{\psi}])$ is actually trivial because its surface anomaly $[\tilde{\psi}]$ can be realized as a gapped symmetric state.

As an example, we apply this bulk-boundary correspondence again to the first example in the section "Second-page no-open-edge conditions". The $L$-isomorphism maps the relation in Eq. (43) to the following bubbling relation,

$$d_r L([\psi]) = L([\tilde{\psi}]). \tag{59}$$

This indicates that $L([\tilde{\psi}])$ is a 3D TCS trivialized by the second-page bubbling process $L([\psi])$. This provides an explicit example illustrating that it is necessary to consider higher-dimensional trivialization processes as discussed in the section "Further considerations", which can be computed using the higher-page $d_r$ maps introduced in the section "Second-page bubbling equivalences".

## Data availability
Data of bosonic-SPT classification protected by 2D wallpaper groups and 3D space groups together with various onsite symmetry groups can be found in Sec. VII of SI. Other data that support the findings of this study are available from the corresponding author upon request.

## Code availability
All numerical codes in this paper are available upon request to the authors.

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

## Acknowledgements

Y.Q. is grateful to Zheng-Cheng Gu, Zheng-Xin Liu, Shenghai Jiang and Ying Ran for invaluable discussions. Y.Q. also thanks Aspen Center for Physics for hospitality, where part of this work was performed. S.Z.D. and C.F. acknowledge support from Minstry of Science and Technology of China under grant numbers 2016YFA0302400, 2016YFA0300600, from National Science Foundation of China under grant number 11674370, and from Chinese Academy of Sciences under grant number XXH13506-202. Y.Q. acknowledges support from Minstry of Science and Technology of China under grant numbers 2015CB921700, and from National Science Foundation of China under grant number 11874115.

## Author contributions

C.F. and Y.Q. conceived the project and developed the theoretical ideas. Y.Q. derived the spectral-sequence formulas. Z.S. implemented the algorithm and obtained all classification. All authors contributed to the writing of the paper.

## Competing interests

The authors declare no competing interests.
