## [Peer Review File · Nature Communications]

Reviewers' Comments:

Reviewer #1:

Remarks to the Author:

Topological phases protected by spatial symmetry is recently one of the most active topics in condensed matter theory. This article provides a general recipe to classify topological symmetry with spatial and internal symmetry together for both fermionic and bosonic systems. In my opinion, it is an important development in the field and therefore deserves publication in nature communication. That being said, I have to also say that the article is quite technical in nature. Therefore, I have a few minor comments on the presentation of the paper.

1. In Fig. 2(b), the example given here, does it assume periodic boundary condition?
2. In the HOLS M section, the authors first give some pictures about LSM of translational symmetry and projective rep. Later (on page 15), it is generalized to other symmetries. Can the authors give an example for a more general case beyond just translation?
3. My impression after reading is that the general strategy for classifying boson and fermion TCS are the same. The only difference is just that one needs to decorate boson/fermion SPT on the p-blocks for bosonic/fermionic TCS respectively? Is this statement correct? Can the authors stress more about which part of the classification scheme is general for both boson and fermion systems and what is the essential difference between them?
4. On page 13, the authors discuss an example that is beyond layer constructions. The authors state that "Therefore the classification of bosonic TCS in this case is identical with classification of free fermion topological crystalline insulator." I cannot quite make sense of this statement. It would be great if the authors can explain a little on this. It would also be clearer if the authors provide this example beyond layer construction explicitly.

Overall, I think this paper has many interesting and original results and should be published in nature communication after minor adjustments.

Reviewer #2:

Remarks to the Author:

The authors claim that they present a unified scheme for constructing all topological crystalline states, bosonic and fermionic, free and interacting. The key idea is to use the so-called real-space "building blocks" and "connectors". Building blocks are finite-size pieces of lower dimensional topological states protected by onsite symmetries alone, and connectors are "glue" that complete the open edges shared by two or multiple pieces of building blocks. Then they applied this scheme to obtaining the full classification of bosonic topological crystalline states protected by several onsite symmetry groups and each of the 17 wallpaper groups in two dimensions and 230 space groups in three dimensions. They further claim that their real-space construction gives the complete set of topological crystalline states for bosons and fermions. I found that the main results of this manuscript are very interesting, and it warrant a publication in some form. However, I also feel that the presentations are too mathematical, and some claims are too "big".

(a) Does the arrow in Eq. (2) means an exact sequence? If it is indeed the case, please carefully explain the notation and explain why such a mathematical notation is necessary for understanding this physical problem.

(b) The terminology "page" below Eq.(2) is borrowed from the mathematical concept of spectrum sequence. Such a word could be very strange for readers who are not aware of spectrum sequence theory. It is better to start from a physical example or physical picture for the construction instead

of heavy mathematical notation. Necessary mathematical background could be put into appendix or supplementary material. I believe many readers could get lost once they hear about "pages".

(c) Although the results are well tested for interacting bosonic systems as well as some free fermion systems, there is no single example of interacting fermion system. In particular, I believe that the "obstruction" and "trivialization" conditions are much more complicated for interacting systems, provided that the classification scheme for on-site symmetry are already very complicated in interacting fermion systems. Therefore, I believe that the statements such as "for constructing all topological crystalline states, bosonic and fermionic, free and interacting" and "real-space construction gives the complete set of topological crystalline states for bosons and fermions" might be over claimed.

Reviewer #3:

Remarks to the Author:

Recently, there have been quick developments in the understanding of crystalline symmetry protected topological (SPT) phases. One of the systematic approach, which is usually called "block state" or "topological crystal" picture, conjectures that crystalline SPT phases are adiabatically connected to stacks of lower-dimensional SPT phases arranged in some pattern respecting crystalline symmetry. These lower-dimensional SPT phases should satisfy a "gluing condition" such that all the gapless modes in the bulk should be glued together and gapped out while preserving symmetry. Since there are many block states which can be deformed into each other, one then needs to consider all possible deformations as equivalence relations in order to obtain the classifications.

The original proposal is based on physical arguments, and the formal mathematical structure was not clear at that time. There are three papers that appeared around the same time which revealed that the relevant mathematical structure should be the spectral sequence in some generalized homology/cohomology, and this paper is one of them (the other two papers are Phys. Rev. B 99, 115116 (2019), arXiv: 1810.00801). Compare to the other two papers, this paper focuses on phases within group cohomology and works through the details of the spectral sequence explicitly. Moreover, the authors show that many of the computation in the spectral sequence can be made algorithmic. For example, the authors show that there is an explicit formula of "gluing" for phases within group cohomology. Such computations had to be done by considering physical models and see how to glue these models by adding symmetry allowed couplings. With this formula of "gluing", the calculation is significantly simplified and can be made automated, which is done in this paper for bosonic crystalline SPT phases with several internal symmetry groups and all the wallpaper groups and all the space groups. The authors also discuss an application of this approach: deriving generalized Lieb-Schultz-Mattis constraints, which stating that, in some situations, a symmetric, gapped, unique ground state must be a non-trivial SPT state. This paper is also well-written and easy to follow. Readers who are less comfortable with formal mathematics might find this paper more accessible. I do not have particular concern for this paper. I recommend this paper to be published in Nature Communications.

Reviewers' comments:

Reviewer #1 (Remarks to the Author):

Reviewers' comment:

Topological phases protected by spatial symmetry is recently one of the most active topics in condensed matter theory. This article provides a general recipe to classify topological symmetry with spatial and internal symmetry together for both fermionic and bosonic systems. In my opinion, it is an important development in the field and therefore deserves publication in nature communication. That being said, I have to also say that the article is quite technical in nature. Therefore, I have a few minor comments on the presentation of the paper.

Our reply:

The authors are pleased to see that the value of the work has been highly appreciated by an expert. The reviewer raises minor concern for its technicality and presentation, and gives useful comments and questions to help us improve the paper. Each comment is fully addressed below, and corresponding changes are implemented in the revision.

Reviewers' comment:

1. In Fig. 2(b), the example given here, does it assume periodic boundary condition?

Our reply:

In this example, consistent with the constructions elsewhere in this paper, we consider an infinite system with translation symmetries. As a result, decorations in all unit cells must be identical due to the symmetry constraints [see Eq. (4)]. Hence, we only draw one unit cell in Fig. 2(b).

Reviewers' comment:

2. In the HOLSM section, the authors first give some pictures about LSM of translational symmetry and projective rep. Later (on page 15), it is generalized to other symmetries. Can the authors give an example for a more general case beyond just translation?

Our reply:

We thank the reviewer for this interesting question. More examples of LSM-type constraints beyond just translation symmetries are discussed in two papers that appeared after this work: arXiv:1907.08204 by D. Else and R. Thorngren and arXiv:1907.08596 by S. Jiang, M. Cheng, Y. Qi and Y. Lu. In particular, several explicit examples are given in Sec. IV of the second paper. These recent progresses also demonstrate that our framework can be used to derive more general LSM constraints for symmetry groups beyond just translation. However, to avoid unnecessary overlap with these works, at this point, we do not want to include more examples in our paper. In the revised manuscript, we added a reference to the aforementioned recent paper by Else and Thorngren. (This paper is cited because we had some private communications with its authors when we are writing ours.)

Reviewer's comment:

3. My impression after reading is that the general strategy for classifying boson and fermion TCS are the same. The only difference is just that one

needs to decorate boson/fermion SPT on the p-blocks for bosonic/fermionic TCS respectively? Is this statement correct? Can the authors stress more about which part of the classification scheme is general for both boson and fermion systems and what is the essential difference between them?

Our reply:

We thank the reviewer for this important question. The difference between classifying bosonic and fermionic TCS lies in the specific rules for decorating onsite-symmetry SPT states on each p-block, while the form of such rules remains the same for bosons and fermions. In the revised manuscript, the general form of these rules are summarized in Sec. IIB of our paper. Essentially, for a given dimension p and a given symmetry group G , one needs the following rules

1. The classification of SPT phases, which forms an Abelian group. We denote this by $\Phi^p(G)$.

This information is enough to determine the second-page approximation of the classification. However, in order to proceed to compute higher-page approximations, the following additional information is needed:

2. A set of fixed-point symmetric wave functions for SPT phases in each dimension, which we denote by $\Psi^p(G)$, and a coboundary operator representing the bulk-boundary correspondence, which we denote by $d: \Psi^p(G) \rightarrow \Psi^{p+1}(G)$.

We remark that here the terminology 'SPT phases' includes the so-called invertible topological orders, like the Majorana chain in 1d and $(p+i)$ -superconductors in 2d (these are not included as SPT phases in some literatures.)

To better explain these concepts, we provide additional details of these data in the revised manuscript. We added a new subsection, Sec. IIB, to summarize the general forms of rules for decorating different types of SPT phases. For bosonic SPT phases, such rules are reviewed in Appendix A2. For fermionic SPT phases, such rules are outlined in Appendix A3. Furthermore, in Sec IV, we added an example to demonstrate the computation of interacting-fermion SPT phases using real-space recipes following the rules.

Reviewer's comment:

4. On page 13, the authors discuss an example that is beyond layer constructions. The authors state that "Therefore the classification of bosonic TCS in this case is identical with classification of free fermion topological crystalline insulator." I cannot quite make sense of this statement. It would be great if the authors can explain a little on this. It would also be clearer if the authors provide this example beyond layer construction explicitly.

Our reply:

We thank the reviewer for this question. In the original manuscript, the analogy we made between bosonic and free-fermion TCSs is that they have similar mathematical structures: they are both constructed by decorating 2d topological states on 2-cells, and the decorated 2d block states both have a Z_2 classification protected solely by onsite symmetries. Moreover, the anomalies they created on 1-cells also have a Z_2 classification, regardless

of whether the 1-cells have additional point-group symmetries. Therefore, the classification of such TCSs can be stated as the same mathematical problem of finding inequivalent ways of decorating Z_2 numbers on 2-cells, such that the decoration is symmetric with respect to the space-group symmetry, and the Z_2 numbers add to zero on each 1-cell. Of course, the two types of TCSs have very different physical nature, as the decorated root states are totally different: they are the Z_2 bosonic SPT state (the Levin-Gu or the CZX state) and the 2D TI, respectively. So the analogy between the two cases is purely mathematical.

We agree with the reviewer that this analogy can be confusing to the readers without a detailed explanation, and it is also quite technical and does not help much to the understanding of the content of this paper. Therefore, in the revised manuscript, we rewrote the example in Sec. IV. Following the suggestion of the reviewer, we added an explicit construction of the example of bosonic SPT state beyond layer constructions, and we removed the use of the analogy to free-fermion TCIs to avoid confusion. We also added an appendix (Appendix F) to explain the cell structures in Fig. 5.

Reviewer's comment:

Overall, I think this paper has many interesting and original results and should be published in nature communication after minor adjustments.

Our reply:

The authors thank the reviewer for his/her valuable comments/questions that help us improve the presentation, and the favorable assessment.

Reviewer #2 (Remarks to the Author):

Reviewer's comment:

The authors claim that they present a unified scheme for constructing all topological crystalline states, bosonic and fermionic, free and interacting. The key idea is to use the so-called real-space "building blocks" and "connectors". Building blocks are finite-size pieces of lower dimensional topological states protected by onsite symmetries alone, and connectors are "glue" that complete the open edges shared by two or multiple pieces of building blocks. Then they applied this scheme to obtaining the full classification of bosonic topological crystalline states protected by several onsite symmetry groups and each of the 17 wallpaper groups in two dimensions and 230 space groups in three dimensions. They further claim that their real-space construction gives the complete set of topological crystalline states for bosons and fermions. I found that the main results of this manuscript are very interesting, and it warrant a publication in some form. However, I also feel that the presentations are too mathematical, and some claims are too "big".

Our reply:

The authors are grateful for the recommendation as well as the criticism offered by this reviewer. This reviewer finds our results interesting, while raising the issue that the presentations are "too mathematical" and that some scientific claim is insufficiently supported by the content. To fully address the criticism, we made significant modification to our presentation at many places, and we also clarified the application of the theoretical framework, as well as the technical difficulty that has so far

been hindering a full classification for interacting fermions.

Reviewer's comment:

(a) Does the arrow in Eq. (2) mean an exact sequence? If it is indeed the case, please carefully explain the notation and explain why such a mathematical notation is necessary for understanding this physical problem.

Our reply:

We thank the reviewer for this important question. Indeed, the sequence in Eq. (2) is exact, but this fact is not obvious, nor is it implied by this notation, and it should be clearly pointed out.

Here, in Eq. (2), we write down the so-called chain complex of the CW-complex Y , which is a widely used mathematical object in algebraic topology.

In general, the chain complex for an arbitrary topological space is not exact, and its nonexactness is measured by the homology groups of the topological space. However, in our case, the CW-complex Y has trivial homology groups, because it is the space R^3 , which is contractible. Therefore, in fact, the sequence in Eq. (2) is exact.

In the revised manuscript, we added a short explanation below Eq. (2) to explain its exactness and the reason behind it.

Reviewer's comment:

(b) The terminology "page" below Eq. (2) is borrowed from the mathematical concept of spectrum sequence. Such a word could be very strange for readers who are not aware of spectrum sequence theory. It is better to start from a physical example or physical picture for the construction instead of heavy mathematical notation. Necessary mathematical background could be put into appendix or supplementary material. I believe many readers could get lost once they hear about "pages".

Our reply: We thank the reviewer for raising this important issue. In general, we think the concept of "pages" is an essential ingredient in our framework of computing space-group SPTs, and it needs to be properly introduced to the readers for them to understand our paper. Therefore, we do not think avoiding the terminology "pages" is a good way. (In particular, we think replacing the word "pages" with another word only adds confusion.) In fact, in our manuscript, we introduced the terminology "pages" in introduction (see the fifth paragraph in Sec. I.) Realizing that the title of Sec. IIC might be confusing for readers who skipped our introduction, we changed the title to "'First page' (first-order approximation) candidates for TCS" to give a short explanation of the terminology. Moreover, we add one sentence to the introduction to further explain the terminology.

Reviewer's comment:

(c) Although the results are well tested for interacting bosonic systems as well as some free fermion systems, there is no single example of interacting fermion system. In particular, I believe that the "obstruction" and "trivialization" conditions are much more complicated for interacting systems, provided that the classification scheme for on-site symmetry are already very complicated in interacting fermion systems. Therefore, I believe that the statements such as "for constructing all topological

crystalline states, bosonic and fermionic, free and interacting" and "real-space construction gives the complete set of topological crystalline states for bosons and fermions" might be over claimed.

Our reply:

We thank the reviewer for this important comment, and we agree with the reviewer that the computation for interacting-fermion systems are much more complicated than for bosonic and free-fermion systems, because of the rules for onsite SPTs are much more complicated. However, the focus of our work is to build crystalline SPTs using a real-space construction, assuming that the rules for onsite SPTs are known. The rules of real-space constructions, including the no-open-edge conditions and the bubble equivalences, are quite generic and only depends on some abstract forms of the classification of onsite SPTs of the corresponding type. To be more specific, the real-space construction depends on the following information:

1. The classification of SPT phases, which forms an Abelian group. We denote this by $\Phi^p(G)$.

This information is enough to determine the second-page approximation of the classification. However, in order to proceed to compute higher-page approximations, the following additional information is needed:

2. A set of fixed-point symmetric wave functions for SPT phases in each dimension, which we denote by $\Psi^p(G)$, and a coboundary operator representing the bulk-boundary correspondence, which we denote by $d:\Psi^p(G)\rightarrow\Psi^{p+1}(G)$.

To better explain these concepts, we provide additional details of these information in the revised manuscript. We added a new subsection, Sec. IIB, to summarize the general forms of rules for onsite SPTs that are required for our real-space construction. For bosonic SPT phases, such rules are reviewed in Appendix A2. For interacting-fermion SPT phases, such rules are outlined in Appendix A3. Furthermore, in Sec IV, we added an example to demonstrate the computation of interacting-fermion SPT phases using real-space recipes.

Our real-space construction, including the no-open-edge conditions and the bubbling equivalences, is then expressed in terms of $\Phi^p(G)$ and $\Psi^p(G)$, and therefore independent of the details of the onsite SPT classifications. Consequently, our construction is generic and should in principle apply to interacting-fermion systems. However, due to the complex nature and the lack of full details of classification schemes for onsite interacting-fermion SPTs, our paper does not contain a systematic and fully automated computation of interacting-fermion SPT classifications.

Reviewer #3 (Remarks to the Author):

Reviewer's remark:

Recently, there have been quick developments in the understanding of crystalline symmetry protected topological (SPT) phases. One of the systematic approach, which is usually called "block state" or "topological crystal" picture, conjectures that crystalline SPT phases are adiabatically connected to stacks of lower-dimensional SPT phases arranged in some pattern respecting crystalline symmetry. These lower-

dimensional SPT phases should satisfy a "gluing condition" such that all the gapless modes in the bulk should be glued together and gapped out while preserving symmetry. Since there are many block states which can be deformed into each other, one then needs to consider all possible deformations as equivalence relations in order to obtain the classifications.

The original proposal is based on physical arguments, and the formal mathematical structure was not clear at that time. There are three papers that appeared around the same time which revealed that the relevant mathematical structure should be the spectral sequence in some generalized homology/cohomology, and this paper is one of them (the other two papers are Phys. Rev. B 99, 115116 (2019), arXiv: 1810.00801). Compare to the other two papers, this paper focuses on phases within group cohomology and works through the details of the spectral sequence explicitly. Moreover, the authors show that many of the computation in the spectral sequence can be made algorithmic. For example, the authors show that there is an explicit formula of "gluing" for phases within group cohomology. Such computations had to be done by considering physical models and see how to glue these models by adding symmetry allowed couplings. With this formula of "gluing", the calculation is significantly simplified and can be made automated, which is done in this paper for bosonic crystalline SPT phases with several internal symmetry groups and all the wallpaper groups and all the space groups. The authors also discuss an application of this approach: deriving generalized Lieb-Schultz-Mattis constraints, which stating that, in some situations, a symmetric, gapped, unique ground state must be a non-trivial SPT state. This paper is also well-written and easy to follow. Readers who are less comfortable with formal mathematics might find this paper more accessible. I do not have particular concern for this paper. I recommend this paper to be published in Nature Communications.

Our reply:

The authors thank the reviewer for recognizing our work among others' independent and simultaneous contribution, while distinguishing it by its style ("Readers who are less comfortable with formal mathematics might find this paper more accessible") and method ("the computation in the spectral sequence can be made algorithmic"). The authors thank the reviewer for the recommendation.

Summary of other changes not directly related to reviewers' comments:

In this revision, in addition to the changes in response to reviewers' comments that are listed above, we also made the following changes to improve the clarity of our manuscript:

1. In Sec. III A 2, we rewrote the free-fermion example using explicit forms of the boundary Hamiltonian of the connectors on 1-cells, instead of the wave function. We also added one more example.
2. In Appendix G, the results of classification are now expressed as direct products of Abelian groups (like $Z_n \times Z_m$), instead of direct sums of Z -modules (like $Z_n \oplus Z_m$).

Reviewers' Comments:

Reviewer #1:

Remarks to the Author:

The authors have carefully responded to all my questions in the previous report and presentation is improved. I would like to recommend the paper to be published in nature communication.

Reviewer #2:

Remarks to the Author:

The authors have addressed all my concerns and the presentation of this manuscript has greatly improved. The result is novel and of great impact in this subfield of symmetry protected topological phase. I strongly recommend publication in Nature Communication.

REVIEWERS' COMMENTS:

Reviewer #1 (Remarks to the Author):

The authors have carefully responded to all my questions in the previous report and presentation is improved. I would like to recommend the paper to be published in nature communication.

Reviewer #2 (Remarks to the Author):

The authors have addressed all my concerns and the presentation of this manuscript has greatly improved. The result is novel and of great impact in this subfield of symmetry protected topological phase. I strongly recommend publication in Nature Communication.

Response to Reviewers' Comments:

The authors are grateful for the recommendations. In the revised manuscript, we made changes to comply with the editorial requests, which we list below.